# Optocapillarity-driven assembly and reconfiguration of liquid crystal polymer actuators

Zhiming Hu [1,2,4], Wei Fang [3,4], Qunyang Li [3], Xi-Qiao Feng [3✉] & Jiu-an Lv [1,2✉]

Realizing programmable assembly and reconfiguration of small objects holds promise for technologically-significant applications in such fields as micromechanical systems, biomedical devices, and metamaterials. Although capillary forces have been successfully explored to assemble objects with specific shapes into ordered structures on the liquid surface, reconfiguring these assembled structures on demand remains a challenge. Here we report a strategy, bioinspired by *Anurida maritima*, to actively reconfigure assembled structures with well-defined selectivity, directionality, robustness, and restorability. This approach, taking advantage of optocapillarity induced by photodeformation of floating liquid crystal polymer actuators, not only achieves programmable and reconfigurable two-dimensional assembly, but also uniquely enables the formation of three-dimensional structures with tunable architectures and topologies across multiple fluid interfaces. This work demonstrates a versatile approach to tailor capillary interaction by optics, as well as a straightforward bottom-up fabrication platform for a wide range of applications.

[1] Key Laboratory of 3D Micro/Nano Fabrication and Characterization of Zhejiang Province, School of Engineering, Westlake University, 18 Shilongshan Road, Hangzhou 310024 Zhejiang Province, China. [2] Institute of Advanced Technology, Westlake Institute for Advanced Study, 18 Shilongshan Road, Hangzhou 310024 Zhejiang Province, China. [3] AML, Department of Engineering Mechanics, and State Key Laboratory of Tribology, Tsinghua University, Beijing 100084, China. [4] These authors contributed equally: Zhiming Hu, Wei Fang. ✉email: fengxq@tsinghua.edu.cn; lvjiuan@westlake.edu.cn

Programmable assembly of small objects on the liquid surface is of paramount significance for numerous perspectives, ranging from bottom-up fabrication of functional materials and devices[1–6] to a fundamental understanding of biological systems[6–8]. Capillarity, which induces interface distortion and generates capillary forces to attract or repel objects nearby[9], provides a powerful means to assemble objects with a variety of geometries across a wide range of length scales[10,11]. Capillary interactions of objects are determined by both chemical compositions and geometric architectures on their surfaces[5]. Tailoring the two factors, floating objects with various shapes can be assembled into linear chains or ordered arrays[12–16]. In most previous studies, however, capillary interactions have been exploited to assemble objects of fixed shapes. Typically, once the objects are placed on the liquid surface, assembly occurs at different locations determined by the initial sites of the dispersed objects[9]. To date, it is yet a challenge to realize programmable assembly and reconfiguration of floating systems, because of the difficulty to dynamically tune their capillary interactions[17]. Recently, the effort has been directed toward developing programmed capillary assembly techniques by using magnetic microrafts[18] or temperature-responsive hydrogels of three-dimensional (3D) shapes featured by gradient swelling[10,19]. However, the former approach cannot independently control each individual object in the assembled structures due to global actuation of a magnetic field[19], while the latter cannot reconfigure the assembled structures, and dramatic temperature change makes it undesirable, for example, in biomedical applications. Reconfigurable and programmable assemblies turn out to be toughly achieved in synthetic and artificial systems.

In this work, we present a bioinspired strategy that enables light-driven reconfigurable assembly of both two- and three-dimensional structures on the fluid interface. The strategy uses controllable capillary menisci and mutual interactions that can be tuned by dynamically changing shapes of floating azobenzene-functionalized liquid crystal polymer (azo-LCP) actuators, and thus achieves the construction of diverse assembled structures with programmable morphologies.

## Results

**Design of optocapillary-driven assembly system.** In nature, many living organisms have highly efficient strategies to utilize capillary forces to achieve fast locomotion and configurable assemblies. For example, aquatic and semiaquatic insects, such as *Anurida maritima* and the larva of *Pyrrhalta*, exhibit remarkable capability to handle capillary interactions through dynamic adjustment of their body postures on the water surface. By bending their bodies, they can modulate the capillary force of menisci to drive their motions and even assemble the individuals into a colony (Fig. 1a, b)[20–22]. Inspired by these biological phenomena, we propose a strategy to construct soft actuators whose geometry and capillary interactions with surroundings can be dynamically adjusted via a straightforward and programmable optical method. To this end, photoresponsive azo-LCPs were used in our work to fabricate rectangular-shaped actuators as unit building elements. The azo-LCP actuators with parallel or splayed alignment have the ability to dynamically and reversibly transform between flat and bending states through asymmetric contraction or expansion arising from photoinduced conformation changes of azobenzene mesogens[23–32].

UV irradiation can induce directional bending of azo-LCP actuators (Supplementary Fig. 2) while visible light irradiation can make the actuators unbending and return to the flat state (photodeformation mechanism is described in Supplementary Note 1). By controlling the light-induced bending, we could gain the desired curvature of menisci at the edges of the actuators and thereby precisely control the magnitude and direction of attractive or repulsive capillary forces between the actuators. Through this method, for instance, six different assemblies (Cases 1-6) can be constructed by two rectangular actuators (Fig. 1c, and Part 1 in Supplementary Movie 1). These assemblies can be divided into three patterns: end-to-end (Cases 1 and 2), side-by-side (Cases 3 and 4), and T-shaped (Cases 5 and 6). When the two actuators are both controlled to bend upwards or downwards, they prefer to assemble either in the manner of end-to-end (Case 1 and 2) or side-by-side assembly (Case 3 and 4), determined by the initial distance between them (Fig. 1d, see Supplementary Note 2 for details). When one actuator bends downwards and the other bends upwards, they would only favor the T-shaped assembly (case 5 or 6). It's noted that these light-controlled assembling is fast and takes only a few seconds. Also importantly, the optocapillarity-driven reconfiguration can be used to actively control the pattern of assemblies by directionally bending the actuators (Part 2 in Supplementary Movie 1). As shown in Fig. 1e, the three patterns of assembly (end to end, T-shaped, and side by side) can be easily manipulated to reversibly transition to each other.

**Mechanisms of the assembly and reconfiguration.** The optocapillarity-driven assembly can be interpreted by considering the opto-deformation of the actuators and the capillary forces of menisci. The azo-LCP actuator is bent due to the gradient of photo-induced strains along the thickness direction (Supplementary Note 3). After light irradiation, the actuators can keep their stable bending shapes. The bending morphology of the actuators deforms the surrounding menisci, which, in turn, drives the actuators to move, as shown in Fig. 2a. Using the finite element method, we numerically visualize the contours of Gibbs free energies in the systems with one (Supplementary Fig. 7, and Supplementary Note 4), or two actuators (Supplementary Fig. 9, and Supplementary Note 5).

To further reveal the proposed approach, we analyze the assembly of two actuators. The variations in the total Gibbs free energies with respect to the center-to-center distance $d$ and the orientation angle $\alpha$ between the two actuators are given in Fig. 2b–e. It can be seen from Fig. 2b–d that for the two actuators with positive curvature ($c_1$, $c_2 > 0$), the system has two stable assembled morphologies, end-to-end (Fig. 2c) and side-by-side (Fig. 2d). To lower the total Gibbs free energy in the system, both of the actuators will approach and rotate, leading to the end-to-end (Case 1, $d = l$, $\alpha = 0°$, with $l$ being the length of each actuator) or side-by-side assembly (Case 3, $d = w$, $\alpha = 180°$, with $w$ being the actuator width). The free energy value is maximal at $\alpha = 60°$, indicating a potential energy barrier between the two stable states, as shown in Fig. 2b. In addition, the systems consisting of two actuators of other shapes are also analyzed. Two actuators with both negative curvature ($c_1$, $c_2 < 0$), have two similar stable configurations: end-to-end (Case 2) and side-by-side (Case 4). However, the T-shaped assembly is energetically favorable for two actuators with opposite curvatures ($c_1 > 0$, $c_2 < 0$), corresponding to two stable configurations (Case 5 and 6), as shown in Fig. 2e. The choice of the two types of T-shaped assembly depends on the initial relative positions and external disturbances of the actuators. Thus, both experiments and numerical simulations demonstrate that the six cases of stable and equilibrated assemblies shown in Fig. 2f can be easily formed by two actuators.

It is noteworthy that the assembly can not only be well controlled into different stable cases but also can switch reversibly among tunable quadrupolar structures through programming the opto-deformation of the actuators. A flat actuator only induces

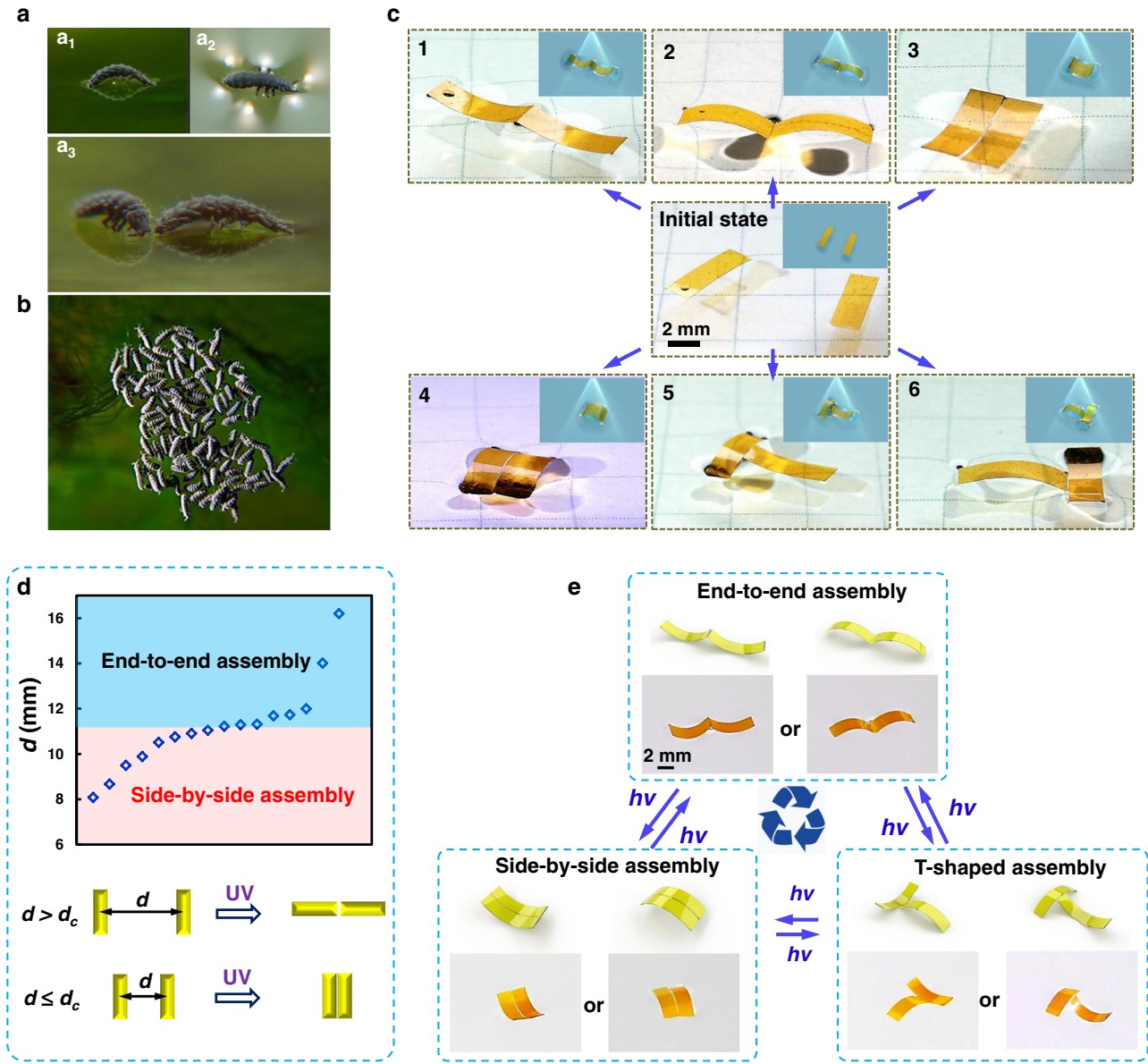

**Fig. 1 Bio-inspired assembly on the water surface.** (**a**, **b**) Capillary force-driven assembly of Anurida into different configurators on the water surface. An anurida arches its back and locks itself into a fixed posture to distort the water surface (a1: lateral view, a2: top view). Anurida generate torques that twist and align them and forces that draw them together (a3). Such forces are in use of generating colonies of hundreds of individuals shown in the below image. Image courtesy of David L. Hu. (**c**) Optocapillarity-driven assembly of two shape-programmed actuators. Photographs showing the six possibilities of favorable assembly of two actuators when transforming from flat to bending state upon the irradiation of 365-nm light (Part 1 of Supplementary Movie 1). The insets schematically illustrate the optocapillarity-driven assemblies. (**d**) Diagram showing the assembly states determined by the initial distance between two actuators. Schematics below the diagram showing the states of two actuators before and after the assembly. $d_c$ is the critical distance that determines the assembly states. When the initial distance (**d**) between two actuators is greater than $d_c$, they tend to form end to end assembly, while d is equal to or less than $d_c$, the side-by-side assembly would occur. (**e**) Optocapillarity-driven reconfiguration of the assembly of two shape-programmed actuators. The reversible transition among three patterns (end-to-end, T-shaped, and side-by-side) can be manipulated through light-driven directional bending of actuators (Parts 2-1 and 2-2 of Supplementary Movie 1). In each state enclosed by the dashed rectangle, the top schematics show the assembly states of the two actuators, and the images below the schematics are the corresponding experimental photographs. hv means the operation of the light. The intensity of UV light is 175 mW cm$^{-2}$; the size of the actuator is 6 mm × 2 mm × 0.03 mm.

shallow water menisci compared to a bent one, corresponding to small capillary forces (Fig. 3a). When the top surface of the actuator is irradiated with UV light, its up-bending dynamically deforms the water surface, with the curvature increasing from 0 to 20 m$^{-1}$. When the up-bending actuator is subjected to irradiation on the bottom surface, both the deformation field in the structure, and the shape of the surrounding menisci can be reversed. The actuator's curvature can be easily tuned by varying the intensity

and irradiation time of actinic light (Supplementary Fig. 10). With an increase in the light intensity from 0 to 220 mW cm$^{-2}$, the curvature changes from 0 to 37.61 m$^{-1}$. Taking advantage of the light-driven transition of the deformation modes, we can actively tailor the morphology of assembly.

In what follows, we investigate how to achieve a desired transformation among the six different assemblies. As a potential energy barrier existing between Cases 1 and 3 hinders their

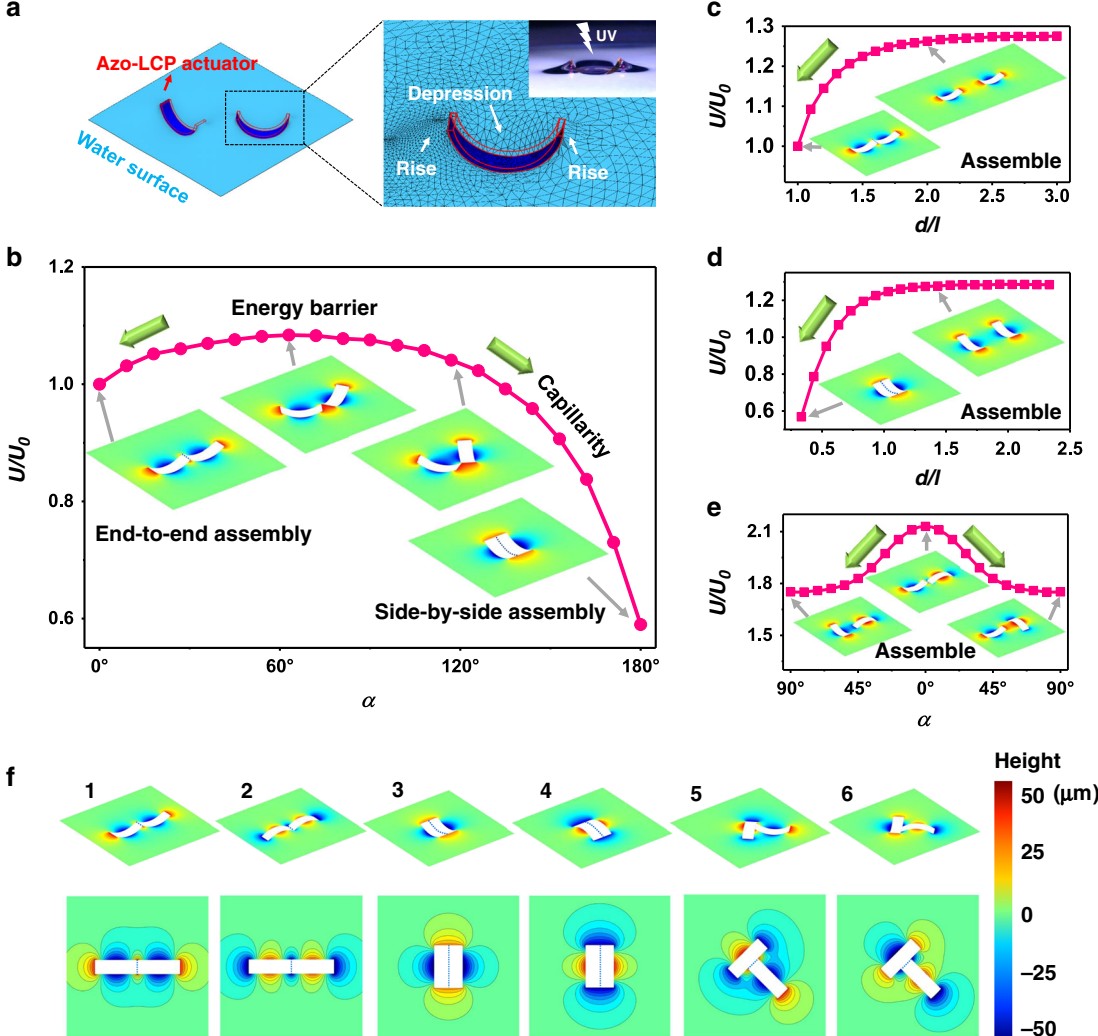

**Fig. 2 Mechanism of optocapillarity-driven assembly.** (**a**) Three-dimensional diagram showing the air-water interface deformation generated by light-induced bending of two azo-LCP actuators. The right image is a schematic diagram of the finite element method. The inset is an optical photograph showing the air-water interface deformation generated by light-induced bending of an azo-LCP actuator. The interface rose near the ends of the bent actuator and depressed in the middle when it bent upwards. The deformed interface adopted a quadrupolar structure. (**b**-**e**) The total Gibbs free energy of the system composed of two actuators varies with the distance $d$ (**c**, **d**) and the orientation angle $\alpha$ (**b**, **e**) in different conditions. As a reference, $U_0$ is the total Gibbs free energy of the system with two actuators end to end, i.e., Case 1. (**b**) The end of two actuators are always in contact at the peak-point during the rotation and $c_1 = c_2 = 20$ m$^{-1}$. (**c**, **d**) The horizontal axis denotes the ratio of the center-to-center distance $d$ to the initial length of the actuator $l$. Here the orientation angle between two actuators is $\alpha = 0°$ (**c**) and $\alpha = 180°$ (**d**) respectively, the curvatures are set as $c_1 = c_2 = 20$ m$^{-1}$. (**e**) The center-to-center distance is fixed as $d = 1.2$ $l$, the curvatures are set as $c_1 = 20$ m$^{-1}$, $c_2 = -20$ m$^{-1}$. The green arrow represents the direction of capillarity in which the potential energy of the system decreases. The inset illustrations show the air-water surface morphology obtained by numerical calculation. The color bar represents the height from the horizontal plane, the same as in (**f**). (**f**) Three-dimensional air-water interface morphology and the corresponding iso-height contour diagrams of six typical stable assemblies of two actuators (Case 1, $c_1 = c_2 = 20$ m$^{-1}$; Case 2, $c_1 = c_2 = -20$ m$^{-1}$; Case 3, $c_1 = c_2 = 20$ m$^{-1}$; Case 4, $c_1 = c_2 = -20$ m$^{-1}$; Case 5, $c_1 = -20$ m$^{-1}$, $c_2 = 20$ m$^{-1}$; Case 6, $c_1 = 20$ m$^{-1}$, $c_2 = -20$ m$^{-1}$).

spontaneous transformation, we have created a versatile optoca-pillary approach to drive such transformation. As shown in Fig. 3b and Supplementary Fig. 11, for a new system where one actuator is positive in curvature and the other is zero ($c_1 > 0$, $c_2 = 0$), if they are initially aligned, they will spontaneously disassemble and rotate locally due to the capillary repulsive force and the torque (Supplementary Fig. 12). The two actuators tend to form a V-shaped configuration when $c_1$ is small ($c_1 = 20$ m$^{-1}$), and a larger $c_1$ would cause a greater repulsive which tends to detach the two actuators (forces analysis is given in Supplementary Discussion). This means that the assembled system can be controlled to reform by adjusting the shape of only one actuator. Thus, we propose an optocapillary route to realize the directional self-assembly of the

actuators by programmable manipulate of light and capillary forces. As indicated in Fig. 3c, since the assemblies **I'** ($c_1 = 20$ m$^{-1}$, $c_2 = 0$, $\alpha = 0°$) and **II'** ($c_1 = c_2 = 20$ m$^{-1}$, $\alpha = 120°$) are unstable, transformations from assembly **I'** to **II** ($c_1 = 20$ m$^{-1}$, $c_2 = 0$, $\alpha = 120°$) and from assembly **II'** to **III** (Case 3, $c_1 = c_2 = 20$ m$^{-1}$, $\alpha = 180°$) can occur spontaneously due to capillary interaction. Importantly, we can easily control the curvature of the actuators via light irradiation in order to achieve the transformation from assembly **I** (Case 1, $c_1 = c_2 = 20$ m$^{-1}$, $\alpha = 0°$) to **I'**, and that from assembly **II** to **II'**. Hence through the coordination of light and capillary forces, the directional transformation from Case 1 to 3, can be realized along the path **I – I' – II – II' – III** (Fig. 3d). Owing to the feasibility of the optocapillary manipulation, which

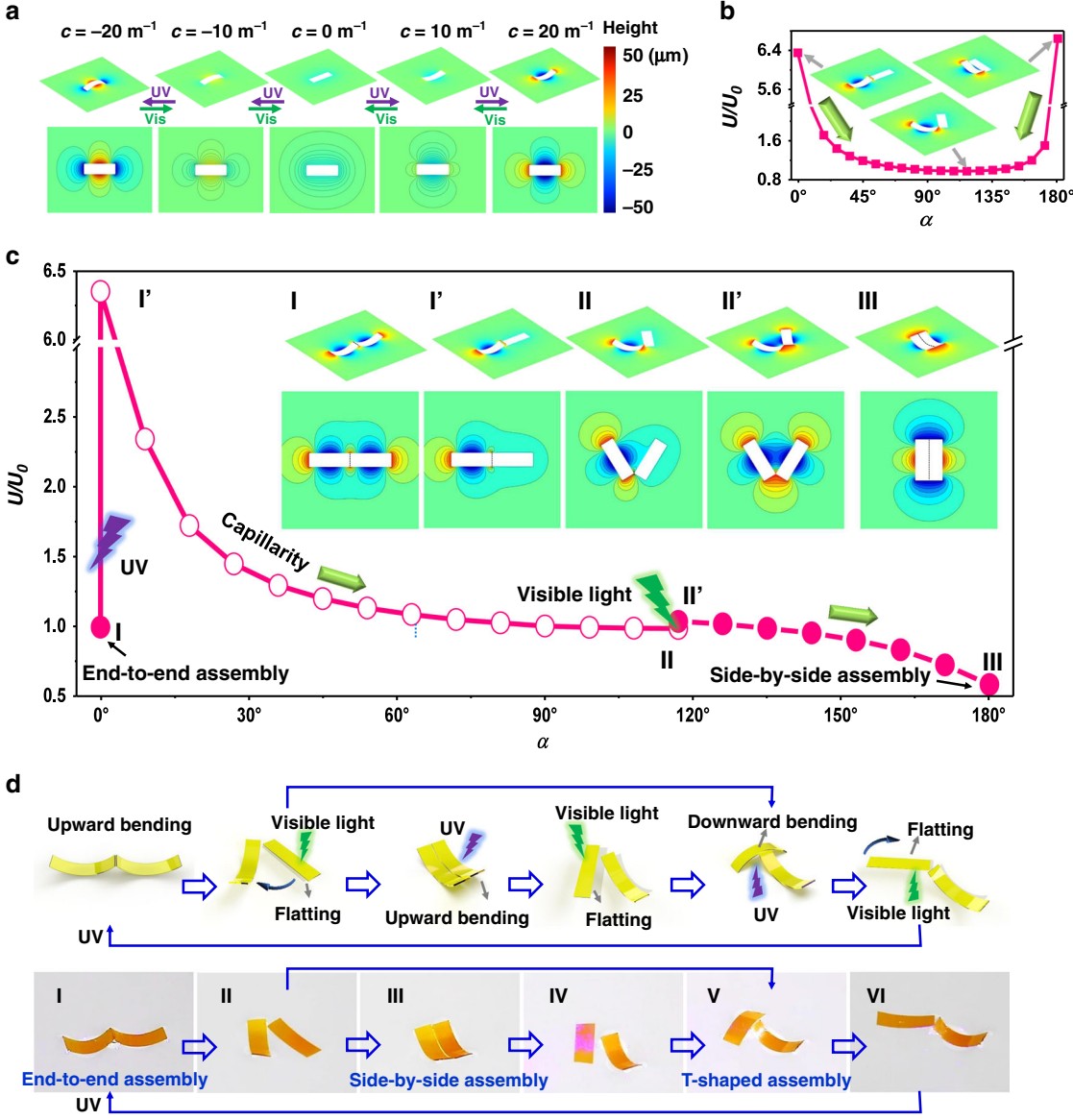

**Fig. 3 Mechanism of optocapillarity-driven reconfiguration. (a)** Three-dimensional air-water interface morphology as well as corresponding iso-height contour diagrams vary when the actuator bends and unbends by alternative irradiation of UV and visible light. The colorbar represents the height from the horizontal plane, same as in (**b**, **c**) (**b**) The total Gibbs free energy of the system composed of an up-bending actuator ($c_1 = 20$ m$^{-1}$) and a flat actuator ($c_2 = 0$) varies with the orientation angle $\alpha$. Light irradiation makes one actuator flat and the other remains curved, when they contact end to end or side by side, the system is extremely unstable and then they rotate spontaneously via capillary to reach equilibrium to avoid assembling in alignment. The arrow represents the direction in which the potential energy of the system decreases. The inset illustrations show the water-air surface morphology obtained by numerical calculation. (**c**) Plot showing the total Gibbs free energy in the two-actuator system during the transformation from Case 1. The curve shows the path to realize the desired transformation and the inset illustrations show the 2D and 3D air-water surface morphology obtained by numerical calculation. (**d**) Schematics (above) and experimental photographs (below) showing optocapillarity-driven reconfiguration of two actuators (Part 2-1 of Supplementary Movie 1). To achieve the reconfiguration from end-to-end assembly (**I**) to side-by-side assembly (**III**), firstly we can illuminate one actuator with visible light to make it flat, and the flattened actuator rotated (**II**). When the orientation angle of the two actuators was close to 120°, the flattened actuator was illuminated with UV light to bend it upwards, and then the two actuators formed side-by-side assembly (**III**). To gain the reconfiguration from side-by-side assembly (**III**) to T-shaped assembly (**V**), one actuator was irradiated with visible light (**IV**). The flattened actuator was irradiated with UV light at the bottom surface to bend it downwards. Then the two actuators aggregated into a T-shaped assembly (**V**). Using the same methods, the T-shaped assembly can be reconfigured again into the end-to-end assembly.

can rotate and move the actuators in a wide range to overcome the energy barriers among the different equilibrium configurations, the path is diversiform and easily programmable. This strategy of assembly is verified by our experiments, as shown in Fig. 3d (Part 2-1 in Supplementary Movie 1). Furthermore, under light driving, one actuator in assembly **III** can be flattened to realize the transformation from **III** to **IV** ($c_1 = 0$, $c_2 = 20$ m$^{-1}$, $\alpha = 120°$)

which will eventually transform to the form **V** (Case 5) through downward bending of the flattened actuator. Therefore, the directional transformation from Case 3 to 5 (or 6) can be well controlled along the path **III** – **IV** – **V**. Similarly, many other paths of structural transformations can be easily realized. For example, the **I** – **V** transformation can be gained via two steps. We first irradiate the bottom surface of one actuator in assemble **I**

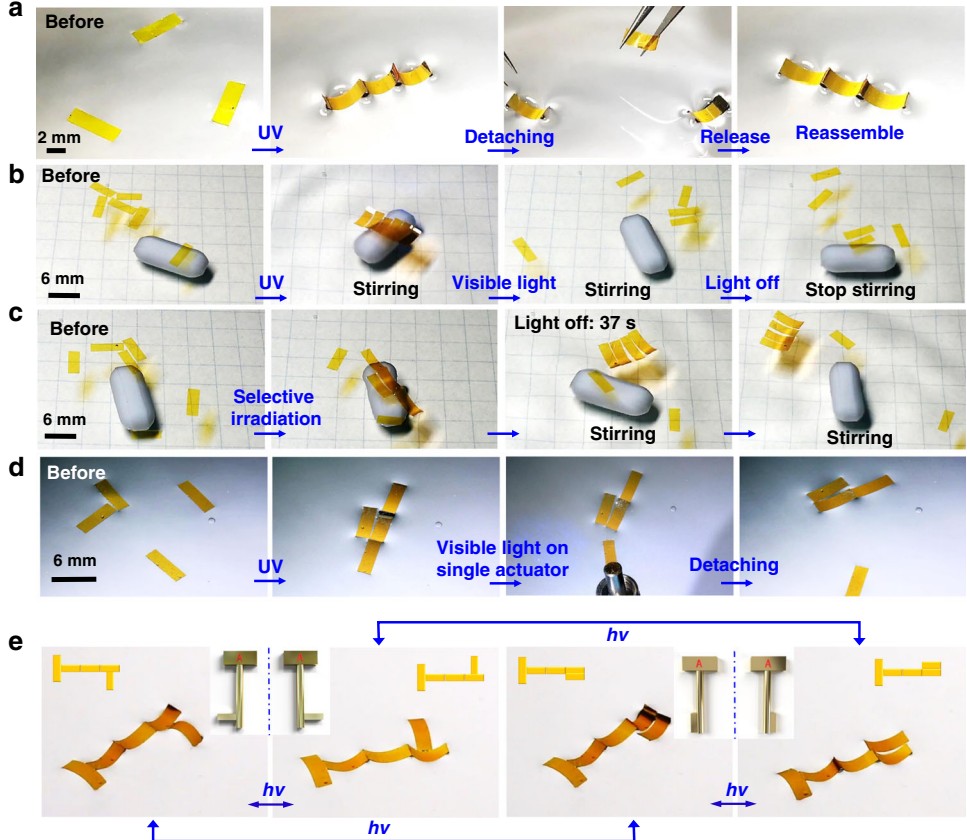

**Fig. 4 Robust and reversible assembly with well-defined selectivity.** (**a**) Time-sequence photographs demonstrate the robustness of the assembled structure. Three actuators assembled into a linear structure after UV irradiation. Two of the three actuators were pulled away to break the linear structure by using two tweezers. After the release of two separate actuators, the three actuators rapidly reassembled into the linear structure (Supplementary Movie 2). (**b**) Time-sequence photographs showing optocapillarity-driven reversible assembly of multiply actuators at air–water interface. At the beginning of the experiment, six actuators were randomly distributed at air–water interface. When the actuators were irradiation with 365-nm light, they aggregated together to form a regular array. Then stirring through a magnetic agitator generates a shearing force that can't break up the aggregation. Switching on 470-nm light, the shearing force immediately splits the array. (Part 1 of Supplementary Movie 3). (**c**) Time-sequence photographs showing selective aggregation controlled by light. There are six actuators randomly dispersed at the interface. Illumination of UV light at any four actuators leads them to aggregate together to form a strong-binding array that enables resist the shearing force induced by the stirring while the rest two actuators without light irradiation are drifted and separated from each other by the stirring (Part 2 of Supplementary Movie 3). (**d**) Time-sequence photographs showing optocapillarity-driven separation of a single actuator from its assembled structure. Four actuators were assembled to form an ordered structure upon the irradiation of 365-nm light. Using 470-nm light to illuminate one of the four actuators can make it to recover the flat state and separate away from the structure (Supplementary Movie 4). (**e**) Photographs showing light-driven switching of the chirality of assembled structures. The key-shaped assemblies formed by 5 actuators possess enantiomers that constitute mirror images. The insets between the adjacent photographs schematically show the two enantiomers in the key-shaped structure (Supplementary Movie 5). Each actuator has a size of 6 mm × 2 mm × 0.03 mm. The intensities of 365-nm and 470-nm light are 175 mW cm$^{-2}$ and 150 mW cm$^{-2}$, respectively. The rotating speed of the magnetic agitator is 200 rpm.

with visible light to make it unbend and flat to form **II**, and then the flattened actuator by UV light is bent downwards to yield the assembly of Case 5 or 6 (direct transition from **II** to **V**). If the two actuators are both bent upwards again through the light manipulation in assembly **V** (Case 5), they will eventually realign as Cases 1 via the two-step path **V** – **VI** – **I**, leading to a complete loop transformation of all reconfigurations. The above experiments show that the light-induced transformations among all assembled structures are reversible, repeatable, and programmable, demonstrating an effective approach to achieve the bottom-up fabrication and reconfiguration of self-assembled structures[17].

**Discussion**. The structures created by optocapillarity-driven assembly are robust to resist external perturbation. As shown in Fig. 4a, a linear structure composed of three actuators, formed via light irradiation, was physically disturbed by manually separating two actuators away. Once the separated actuators have been

released, the three actuators would spontaneously restore the linear structure within a few seconds (Supplementary Movie 2). This reorganization behavior is attributed to an advantage of the proposed assembling technique, which does not need any assistance from external forces. Therefore, the systems have high resistance to external disturbances or damages, can be easily reformed to meet the requirements of new tasks[33]. In addition, the deflection of azo-LCP actuators induced by ultraviolet and visible light has the feature of bistability. The deformed actuators can maintain the bending shape after the light source has been switched off. In other words, the actuators can self-assemble into the desired structure without the need of sustaining irradiation in contrast to some other self-assembly approaches that require continuous irradiation[19]. Furthermore, the photodeformation of azo-LCP actuators has the merit of isothermal control (Supplementary Fig. 16), which is particularly important for biomedical applications.

To better demonstrate controlled reversible assembly, a shearing force generated by stirring the water with a magnetic stirring bar was applied to the actuators. As shown in Fig. 4b, the six rectangular actuators randomly dispersed on the water surface can be assembled together after UV light irradiation. Thereafter we switched on the visible light, it was found that the actuators could restore to the flat state and the capillary interaction among the actuators diminished substantially. As a result, the aggregation was immediately broken by the shearing force, and the actuators were distributed randomly again (Part 1 of Supplementary Movie 3). Therefore, the assembling and disassembling actions can be fully controlled by alternative irradiation of UV and visible light. This provides inspirations to design reassemble or configurational robots or biomedical devices that have high potential for applications in medical and biological engineering[34].

The optocapillarity approach also shows a high ability to manipulate any individual objects within a large ensemble, which is a great challenge for most previously reported techniques of capillary assembly[19]. As shown in Fig. 4c, we irradiate four actuators in a six actuators system with UV light. The irradiated actuators stay together even when subjected to mechanical disturbance, while the other two actuators without irradiation drift away from the structure (Part 2 of Supplementary Movie 3). Similarly, we can easily move a single actuator away from the assembled structure (Fig. 4d, Supplementary Movie 4). These experiments demonstrate a high capability of spatial, temporal, and local manipulation of individual objects in an assembled structure. Furthermore, the optocapillarity approach allows us to easily reverse the handedness of an assembled structure (Fig. 4e, Supplementary Movie 5), indicating an effective strategy for bottom-up, light-driven fabrication of switchable chiral metamaterials.

Furthermore, it is interesting that the actuators possess the functionality of meniscus-climbing like its natural prototype-larva of *Pyrrhalta*. Unlike the previous meniscus-climbers that were limited to only one-way climbing and difficult to adjust the speed and direction of motion[35], the artificial meniscus-climber designed in the present work exhibits reversible meniscus-climbing with controllable velocity and direction. The climber can not only propel itself up a meniscus, but also reversibly slide down, and it even changes its orientation during climbing (Supplementary Fig. 18, Supplementary Movie 6).

In addition, the optocapillarity approach can easily manipulate each building element (actuator) to reconfigure the assembled structures. This is similar to *Anurida maritima* colonies, in which the small insects can directionally bend their bodies to adjust their relative positions, and thus dynamically reconfigure the pattern of the colony in order to adapt to the varying environment. In the artificial systems constructed by the optocapillarity-driven assembling method, each two adjacent actuators can also be treated as a building element with a specific shape (end-to-end, side-by-side, and T-shaped). Through adjusting the assembling states of the building elements, we can actively tune the pattern of an assembled structure. For example, a linear four-actuator system can be reversibly reconfigured into the L or Z shape, with close resemblance to the tetrominoes in the popular video game Tetris, as shown in Fig. 5a.

Upon reconfiguring assembling patterns, the number of the steps for the transformation between different two structures varied, which mainly affected by two factors, namely, the degree of transformation and the number of LCP actuators in the assembling structure. The degree of transformation means the degree of the difference between the initial pattern of the assembling structure and the target pattern of the reconfigured structure. The greater the difference, the higher the transformation degree, and many more steps must be involved to achieve transformation. For example, for the assembling structure composed of 4 actuators, the degree of transformation from pattern 1 to pattern 6 is relatively low (Supplementary Fig. 19c). Because the difference between them is that only two actuators at the end of their structure have a different orientation. And two steps to change the orientation of the actuators at the end of structure 1 is enough to achieve the transformation, which only takes approx. 7 s (Part 2 in Supplementary Movie 7). Whereas for the transformation from pattern 8 to pattern 23 (Supplementary Fig. 19c), the pattern difference between these two structures is relatively high. And it will require multiple steps, and additionally, many parallel paths with a different number of transformation steps exist in this complex transformation. As shown in Part 3 of Supplementary Movie 7, five different parallel paths composed of 3–5 steps take approx. 62 s (average consumed time) to complete the reconfiguration. In other words, this complex transformation will require more optical operations than the simple transformation of pattern 1 to pattern 6. Moreover, if many more actuators were involved in the assembly, the number of parallel paths will dramatically increase. Because, with the capillary forces imposed by multiple polymer actuators, the interactions among actuators become extremely complex and there can be many local energy minima that trap the systems in metastable configurations. It means that more intermediate configurations will emerge, and lead to more optical operations required to achieve the target pattern, especially in the case of the high degree of transformation.

By increasing the number of building elements, we can gain a rich assortment of assembly patterns. Fig. 5b shows that we can construct a diversity of assembled structures with reversible and repeatable patterns. For example, we can sequentially assemble nine rectangular actuators into a large number of different structures, as shown in Fig. 5b (Supplementary Fig. 25 and Supplementary Movie 8). These assemblies and their reconfigurable capability are of great significance for potential applications in the fabrication of structures and devices with tunable mechanical, optical, or electronic properties in the fields that demand dynamic and programmable control of assembling structures, such as robotic swarms, advanced sensor devices, and synthetic swimmers[6,16,19,36,37].

Besides planar structures, the proposed method allows synchronal assembling of actuators at multilayer liquid interfaces (Fig. 6a, Supplementary Movie 9). Even more impressively, it also enables collaborative assembling of the actuators across two adjacent liquid interfaces, and thus realizes the construction of programmed 3D ordered structures (Fig. 6b, Supplementary Movie 10). These 3D synergistic assemblies provide an effective approach to fabricate novel functional structures, hierarchical devices, and architectures, which have great potential for technologically significant applications[38–40].

In summary, we have presented a versatile approach to realize programmable assembly of small floating objects through capillary force arising from light-induced bending of azo-LCP actuators. In contrast to other programmed capillary assembly techniques, this optocapillarity-driven approach shows the advantage to achieve active and programmable reconfiguration of assemblies. It not only enables contactless manipulation of basic building elements (azo LCP soft actuators) to assemble into a diversity of ordered structures, but also allows reversible and repeatable reformation of the assembled structures. We expect it finds use in synthetic robots and swarm robots. For example, the programmable assembly of soft actuators can inspire scientists and engineers to design modular components that enable spontaneously and programmable assembling of

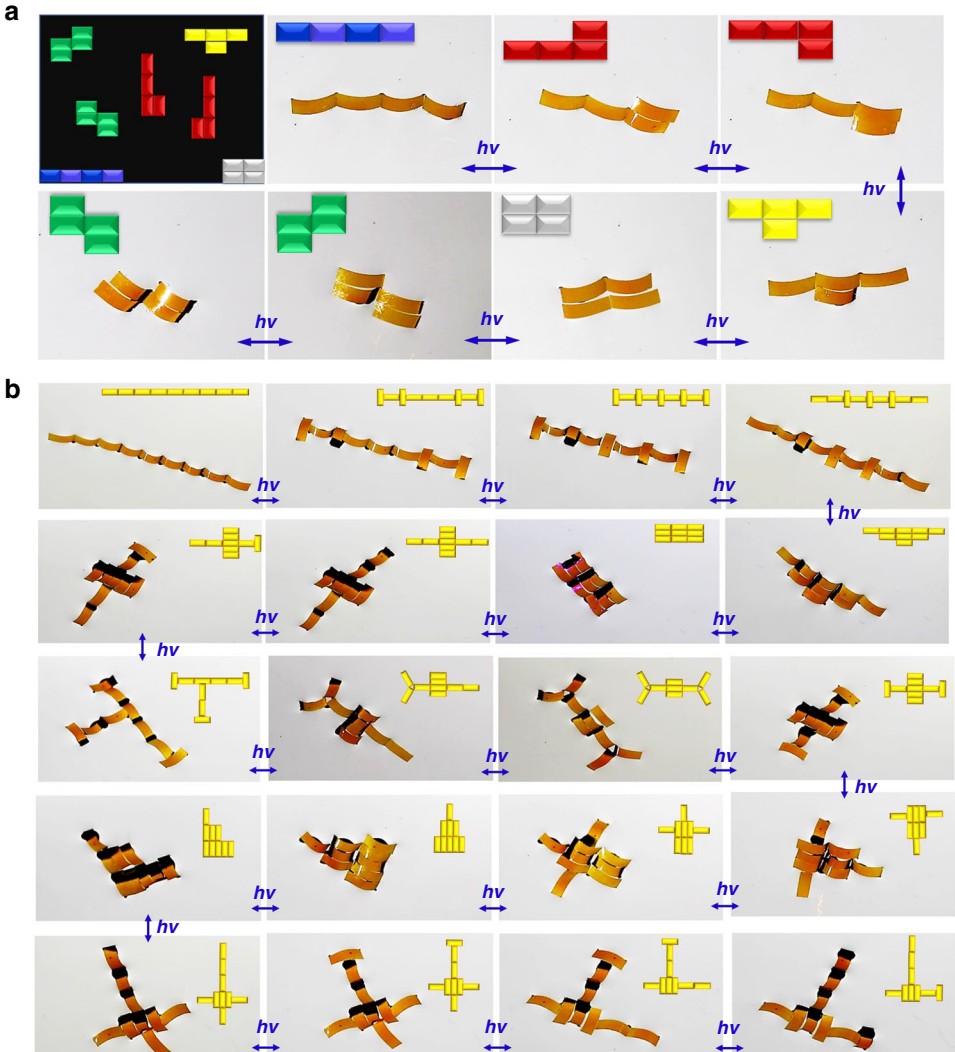

**Fig. 5 Versatility of optocapillarity-driven programmable reconfiguration.** (**a**) Photocontrol of four rectangular actuators to assemble into structures like the tetrominoes of the popular video game Tetris. The L and Z-shaped structures have chirality which can be switched from one enantiomer to another by light-driven reconfiguration. The first image shows the tetrominoes of the video game Tetris (Part 1 of Supplementary Movie 7). (**b**) Photographs showing light-induced reconfiguration of nine rectangular actuators into diverse structures with different patterns. The transformation between any two patterns is achieved by alternative localized irradiation of 450-nm and 360-nm laser as well as modulation of the incident direction of the laser (Supplementary Movie 8). The inset at the top right corner of each photograph schematically indicates the assembled structure. The intensity of the ultraviolet laser and visible laser is 60 mW cm$^{-2}$ and 45 mW cm$^{-2}$, respectively. The size of the rectangular actuators is 6 mm × 2 mm × 0.03 mm.

fully equipped multifunctional robots or devices. The feasibility of optocapillarity-driven dynamic transformation provides a strategy to fulfill reconfigurable designs of swarm robots that could adjust their ensembles into diverse morphologies to adapt to multiple tasks[37]. Most intriguing, our optocapillarity-driven system possesses the unique feature that enables 3D synergistic assemblies at multilayer liquid interfaces, which is only achievable with the ability of photodeformable actuators to achieve positive and negative mean curvature, and have potential application in bioengineering to dynamically construct hierarchical scaffolds for tissue and cell culture. Furthermore, we have unraveled the dynamic and collective responsiveness of the deformable actuators at the liquid interface, and the mechanistic principle of this assembling method ensures that the obtained assemblies can be extended to diverse potential applications. It is anticipated that the optocapillarity-driven assembling strategy will benefit such areas as optomechanical systems, synthetic microrobots, biomedical devices, construction of hierarchical structures and systems.

## Methods

**Preparation of LCN actuators**. The chemical structures of reactive monomers DA11AB6 and C9A (the molar ratio 4:1), and photoinitiator Irgacure 784 (2 wt%) are shown in Fig. S1. The monomers were mixed with the photoinitiator in dried dichloromethane. After being left for approx. 12 h, the dichloromethane was evaporated, and the mixture powder was obtained. Liquid crystal cells with rubbed layers were used for LC alignment. The mixture was filled into the cells by capillary force at 110 °C and then it became an isotropic liquid. Once the cells were totally filled, the temperature was decreased from 110 °C to 95 °C at an annealing speed of 0.1 °C min$^{-1}$. Then the polymerization was performed by irradiating the cells with 550-nm light at 95 °C for approx. 6 h. The light intensity during polymerization was approx. 16 mW cm$^{-2}$. After photopolymerization, the LCN films were peeled off from the cells and cut into the desired shapes by a razor blade.

**Observation of the assemblies and reconfiguration**. The assembling and reconfiguration processed of the actuators on liquid interfaces (photographs and movies) were recorded by a super-resolution digital microscope (Keyence, VHX-1000C) or a digital camera (Canon, EOS 80D(W)). UV light was generated by an LED lamp (Omron ZUV-C30H for $\lambda = 365$ nm) or a Laser source (UV-FN-360-500 for $\lambda = 360$ nm) while visible light at 470 nm was obtained from an LED irradiator (CCS, HLV-24GR-3W) or a laser source (MDL-III-450-50Mw for $\lambda = 470$ nm). The light intensity was monitored by a Newport 1917-R optical

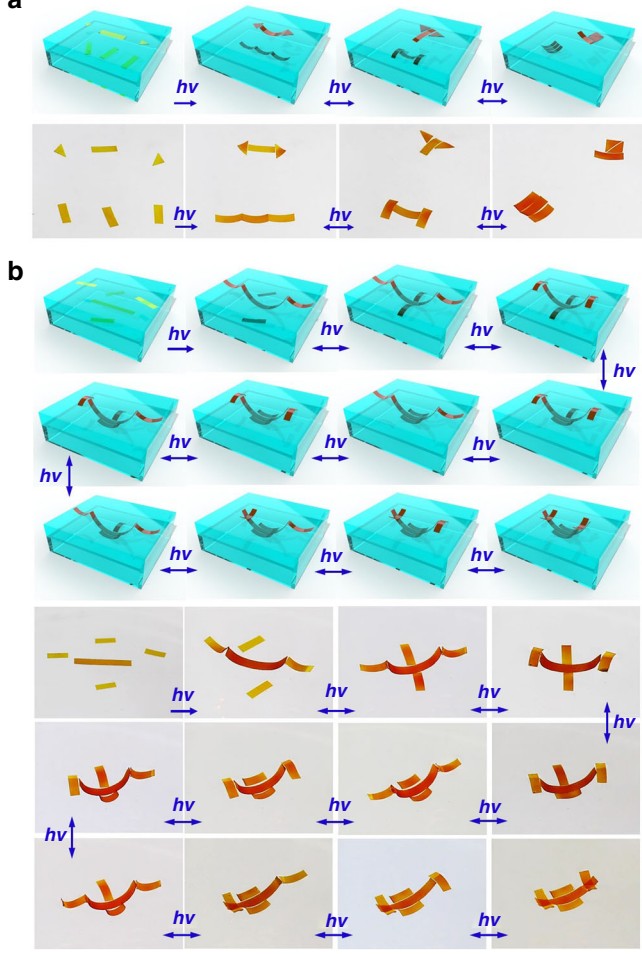

**Fig. 6 Optocapillarity-driven programmable assembly and reconfiguration at multiply fluid interfaces.** (**a**) Schematics (above) and experimental photographs (below) showing optocapillarity-driven assembly at multiply fluid interfaces (Supplementary Movie 9). Two triangular actuators and a rectangular actuator were placed at the air–water interface while three rectangular actuators were placed at the water/FC-70 interface below the air-water interface. (**b**) Schematics (above) and experimental photographs (below) showing the photocontrol of the actuators at spatial interfaces to achieve 3D synergistic assemblies and reconfiguration. Two rectangular actuators were placed at the air-water interface while a longer rectangular actuator and another two rectangular actuators were placed at the water/FC-70 interface below the air-water interface. Upon light irradiation the five actuators assembled into a 3D hierarchical structure (Supplementary Movie 10). The first image in each row of images exhibits the initial location of the actuators before light irradiation. Upon light irradiation, the actuators aggregated and formed regular structures. The transformation of the structures was achieved by the localized irradiation of 450-nm and 360-nm laser with modulation of the incident direction. The intensity of the ultraviolet laser and visible laser is 60 mW cm$^{-2}$ and 45 mW cm$^{-2}$, respectively.

power meter with HIOKI 2018 photodetector or a laser power meter (Merry Change, LP-3).

**Simulation**. A finite-element method was used to simulate the deformation of the actuator and water surface. In the calculation model, the water surface within a region of 6 cm × 6 cm was meshed into more than 200,000 triangular elements in order to ensure the high accuracy of the simulation. The actuators were placed in the central region. To minimize the total Gibbs free energy in the system, a shooting method was adopted. The capillary forces and moments were solved numerically as the partial derivatives of the energy with respect to the displacement

and rotation angle, respectively. The other parameters used in the calculation are given in Supplementary Table 1.

## Data availability

The source data underlying Figs. 1d, 2b–e, 3b, c, Supplementary Figs. 1d, 1e, 7c, 7d, 8d, 8e, 10a, 10b, 11a, 11b, 12c, 12d, and 13 are provided as a Source Data file available in figshare with the identifier [https://doi.org/10.6084/m9.figshare.12910175.v1]. All other data supporting the findings of this study are available from the corresponding authors upon request.

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

## Acknowledgments

This research was supported by the Foundation of Westlake University, National Natural Science Foundation of China (51873197 and 11921002), and 151 Talent Project of Zhejiang Province.

## Author contributions

J.L. and X.Q.F. conceived the research; Z.H. and J.L. designed the experiments; J. L. conceived and designed 3D assemblies across multiple fluid layers; Z.H. performed the experiments; J. L. and Z. H. analyzed the experimental data; W.F., Q.L., and X.Q.F. built the analytical and numerical models. All authors contributed to writing the manuscript.

## Competing interests

The authors declare no competing interests.
