## [Peer Review File · Nature Communications]

Reviewers' comments:

Reviewer #1 (Remarks to the Author):

This paper presents novel and interesting data on the light-driven assembly of polymer sheets that are floating on a liquid. The goal of the paper is to show new configurations of parallel polymer sheets to take up various topological arrangements and to show that the assembly is reversible. The topic is of interest and worth a publication.

The authors have done a remarkable job by recording a multitude of videos to show moving images of the assembly and disassembly steps.

The paper motivates the need for such versatile assembly systems by potential need in wide ranging application, but did not really go into detail about specific needs such as precision, forces, time scales, etc.

The paper generally reads quite well, although there are some language flaws here and there.

The authors motivate their work by listing consistently multiple papers. For instance for general assembly for various areas are listed in [1-8] and would benefit from a more precise justification of the cited papers, one by one.

In line 69, the authors start introducing the material system they are using (azo-LCP).

Line 71, what means 'splay'?

Line 74, lists 23-32 papers on photo-induced changes of this polymer. This number is high and is not justified. Also, nowhere in the paper the authors are discussing the physics behind the opto-capillary effect. There is a nice paper [in International Journal of Solids and Structures 128 (2017) 36-49] that discussed in all detail (notabene: this reviewer has no link to this paper and just found it while searching for some background information on the physics and chemistry of the LCP). The authors of this paper should include some basic information on the mechanism of bending and what forces are involved in the assembly process. It can be done by linking to existing information in other papers already published, to assess the quality of the assembly.

line 89/90: what justifies the number of 9 actuator and 19 structures?

line 102: what means this sentence with "... cutting-edge research fields"

line 105 onward: the explanation of the mechanism is very basic, and more physics should be included to better understand and predict the system

line 117: typo 'low'

line 129 onward: the mechanism of top versus bottom illumination is not well described; maybe a sketch with a drawing and explaining the physico-chemical mechanism would be helpful

line 140: finally some numbers on the bending radius as function of the light source intensity; but what is the rational behind? what is the model and physics? it is just an observation but no explanation

line 207: after assembly, what is the force that keeps them together (and what force is needed to break it appart?_

Videos: they are in general very nice and informative; but maybe there are too many; also it seems that for some assembly task the light source has to be adjusted very close and precise, so one wonders that the assembly could also be done by a tweezer tool; sometimes there is strong light coming in to the camera which is not ideal.
video 4 shows some bubbles; what are they?

figure 1: ia1: later view should be 'lateral view'

figure 2: why these numbers 9 and 19?

The paper currently lacks the physics explanation of the effect, without which it is not yet ready for

publication.

Reviewer #2 (Remarks to the Author):

The paper by Hu et al. "Optocapillarity-driven assembly and reconfiguration of liquid crystal polymer actuators" shows an interesting application of liquid crystal elastomers at the water interface. The elastomer stripes, which can be bent by light due to the presence of azobenzene, change their shape and thus change their capillary interactions. With this stratagem the authors obtain a tunable "cheerios effect" and are able to create reversible assemblies of elastomer stripes (and other shapes) at the interface by regulating the wavelength and the direction of the illumination.

I enjoyed reading this paper very much, and I think this research is interesting and worthwhile. However, I cannot recommend it for publication on Nature Communications for the following reasons.

1) Despite the fact that these experiments have not been done before, to the best of my knowledge, I cannot find anything novel enough in the paper to grant its publication on Nature Communications. Reversible assemblies of soft materials at interface was shown in a system with a richer behavior by Bae et al. *Materials Horizons* 2017 (ref. 10 in the manuscript). The bending of azobenzene-containing elastomers is very well controllable and well known (I realize this is not the main point of the paper). Most importantly, the capillary interactions between objects with the shape of the elastomers have also been extensively characterized (Loudet et al, *PRL* 2005, Lewandovsky et al. *Soft Matter* 2009, Hu & Bush, *Nature* 2005 and others). This paper combines these elements, but I think the novelty is limited and therefore I would recommend the paper for publication in a more materials-focused journal.

2) One of the main claims of the paper is how well controllable this system is. The videos however do not fully support this claim, in my opinion. For example, upon irradiation elastomers seems to fall either in state 1 or 3 almost randomly (side-to-side or head-to-tail assembly). The energy difference between these two states seen in simulations is not supported by a study of how frequently the two modes of assembly are observed. One of the elements which I cannot find in the paper is the importance of the initial relative position of the two elastomers when they are irradiated. This was definitely important in the assembly of anisotropic particles and I expect it to be a relevant parameter also here. The 19 configurations shown in figure 2, likewise, seem quite arbitrary, and from the video they seem to be produced by a series of trials and errors in the assembly, but there is no analysis of this in the paper (for examples, which switches are very favorable and reliable, and which are not?). The supplemental material contains information on the curvature of the elastomer as a function of irradiation time, but an example of how the difference in mean curvature (with the same sign) affects the assembly is missing.

The observation of the system stability under stirring is extremely interesting, but for example I can see that some head-to-tail assembly becomes side-to-side after stirring, a phenomenon which was not explained.

Given these objections I don't recommend publication on this journal but I would definitely recommend publication on a more specialized journal. However, I thought that the final part about the assembly over multiple layers of liquids, and multiple interfaces, was really novel and inspiring. Maybe the authors could further expand on this part and explore in details the capability of this system for 3D assembly, and publish it separately.

Reviewer #3 (Remarks to the Author):

This contribution from Lv and colleagues titled "Optocapillarity-driven assembly and reconfiguration of liquid crystal polymer actuators" is a distinctive contribution that demonstrates the use of photoinduced control of the shape of rectangular elements composed of azobenzene-functionalized liquid crystalline polymer networks affects the wetting of these materials and is a means to optically control assembly via capillarity. Further, the work includes modeling, that

supports the justification of the assembly from a thermodynamics perspective.

In revision or resubmission elsewhere, the authors should:

1 – describe the mechanics of the deformation of the single films more clearly. Although there is considerable literature on the subject, an illustration of the splay geometry and specific discussion of the means by which this enables deformation is important

2 – further, the dynamics of the photoinduced response of these elements is an important consideration and should be discussed. The photographs of the process in the SI are insufficient. How do the materials response over long exposures? Is the deformation stable to continuous exposure of light?

Response to referee 1

Comments 1:

This paper presents novel and interesting data on the light-driven assembly of polymer sheets that are floating on a liquid. The goal of the paper is to show new configurations of parallel polymer sheets to take up various topological arrangements and to show that the assembly is reversible. The topic is of interest and worth a publication. The authors have done a remarkable job by recording a multitude of videos to show moving images of the assembly and disassembly steps.

Response to referee:

We highly appreciate your professional evaluation.

Comments 2:

The paper motivates the need for such versatile assembly systems by potential need in wide ranging application, but did not really go into detail about specific needs such as precision, forces, time scales, etc.

Response to referee:

Thank you for your constructive advice. The work will be improved by conducting more quantitative analysis on precision, forces, time scales, etc. The precision of the assembly is controlled by the bending of actuators. When the bending direction of two actuators is opposite, one bends upwards and the other bends downwards, they always adopt T-shaped assembly. When two actuators have the same bending direction, either bend upwards or downwards. The resulted state of assembly will be affected by the initial distance between them. There is a critical distance that determines assembly states. As shown in Fig. R1, under the premise of radiating light with the same intensity (110 mW cm^{-2}) and placing two actuators in parallel, when the initial distance between two actuators is greater than the critical distance $\sim 11.2 \text{ mm}$, two actuators form end to end assembly, while the distance is less than the critical distance and they would like to aggregate into side-by-side. It should be note that the assembly structure can be precisely tuned by optocapillary-driven reconfiguration after the initial assembly (see Page 5-6 in manuscript for detail).

We have performed force analysis to explain the optocapillary-driven assembly. As shown in Fig. R2a, the forces acting on a single floating actuator can be divided into three parts, which are the gravity mg , the hydrostatic pressure ρgz acting on the lower surface of the actuator, and the surface tension γ acting on the three-phase line (TPL), i.e. the boundary of the actuator. Then, the resultant force on the actuator along the vertical direction and the horizontal direction are respectively

$$\vec{F} = m\vec{g} + \int_{TPL} \vec{\gamma} dl + \iint_{WS} \rho gz \vec{n} ds \quad (r1)$$

$$\vec{M} = \int_{TPL} \vec{L}_p \times \vec{\gamma} dl + \iint_{WS} \rho gz \vec{L}_p \times \vec{n} ds \quad (r2)$$

Here \vec{n} is the local normal vector of the actuator surface, \vec{L}_p is the distance vector from the center of gravity to any point on the TPL or actuator surface. Once the actuator is in equilibrium, the adjacent air-water interface has symmetric geometry (Fig. 3a in the revised Manuscript). When

another actuator approaches, as shown in Fig. R2b, the morphology of air-water interface changes and the symmetry is broken. Therefore, the direction of surface tension acting on the three-phase line varies accordingly, resulting in capillary force F_c and moment M_c . Noted that the contribution of hydrostatic pressure to the horizontal direction and water resistance could be negligible^[1, 2], hence the horizontal motion of the actuator will be dominated by F_c and M_c . The force and moment can be calculated as the partial derivative of energy (Fig. 2b-2e in the revised Manuscript) with respect to the displacement and the orientation angle, respectively. As shown in Fig. R2c, during the end-to-end assembly, the capillary force between two floating actuators is inversely proportional to their distance and possesses a nonlinear characteristic. For two actuators with the same curvature ($c_1 = c_2$), the greater the bending curvature, the stronger the capillary attraction force between them. However, when one of them is treated with visible light and becomes flat ($c_1 > 0, c_2 = 0$), the capillary force is less than zero, which gives rise to a repulsion between them, resulting in the disassembly of the end-to-end state. Furthermore, when this actuator bends in the opposite direction, the capillary repulsion gets stronger to speed up the disassembly process. Fig. R2d shows the capillary moment curve of one actuator with respect to the orientation angle when it is near to another one. It illustrates that the two actuators tend to align parallelly when they are bent in the same direction but vertically when they are not.

For the time scale during assembly, we measured the assembly time and the distance between two approaching actuators upon UV irradiation with different intensities (Fig. R3). With the increase of the radiation intensity from 90 to 170 mW cm⁻², the time required for completing the aggregation decrease from ~ 9 s to ~ 4 s. The moving speed of the two approaching actuators is almost linear in early stage of the assembly, and increases exponentially during the later stage, which is consisted with the growing trend of the capillary attraction force during the approaching.

We have added the abovementioned discussion in section 3 of the revised Supplementary Information.

Figure R1. Diagram showing the assembly states determined by the initial distance between two actuators. Schematics below the diagram showing the states of two actuators before and after the assembly. d_c is the critical distance that determines assembly states. When the initial distance

(d) between two actuators is greater than d_c , two actuators form end to end assembly, while the distance (d) is equal to or less than d_c , they would like to form side-by-side assembly.

Figure R2. Dynamic analysis of the floating actuators. **a.** Force analysis of the single floating actuator. **b.** Capillary interaction between two floating actuators. **c.** Numerical data for capillary force F_c on the actuator (with curvature c_2) vs relative distance d/l from another one (c_1). Here the long axes of the two actuators are in a line, as shown in the inset. **d.** Numerical data for capillary moment M_c on the actuator (c_2) vs the orientation angle α with another one (c_1). Here the center of the c_2 actuator is on the long axis of the c_1 actuator and the center-to-center distance is $d = 1.2l$, as shown in the inset.

Figure R3. Plots showing center-to-center distance between two actuators versus irradiation time when the intensity of UV light source is different. $I_1 = 170 \text{ mW cm}^{-2}$ (blue squares), $I_2 = 155 \text{ mW cm}^{-2}$ (red rhombs), $I_3 = 120 \text{ mW cm}^{-2}$ (green triangles), and $I_4 = 90 \text{ mW cm}^{-2}$ (orange circles). Schematics above showing experiment setup. The initial center-to-center distance of two actuators is fixed at 15 mm, the distance decreases to 6 mm when they finish the end-to-end assembly upon light irradiation. The size of the actuators is $6 \text{ mm} \times 2 \text{ mm} \times 0.03 \text{ mm}$.

[1] Vella, D. & Mahadevan, L. The ‘‘Cheerios effect’’. *Am. J. Phys.* 73, 817-825, (2005).

[2] Yu, Y., Guo, M., Li, X. & Zheng, Q. -S. Meniscus-climbing behavior and its minimum

free-energy mechanism. *Langmuir* **23**, 10546-10550 (2007).

Comments 3:

The paper generally reads quite well, although there are some language flaws here and there.

Response to referee:

Thank you for your comments. We have carefully checked the manuscript and improved the language as much as we can.

Comments 4:

The authors motivate their work by listing consistently multiple papers. For instance for general assembly for various areas are listed in [1-8] and would benefit from a more precise justification of the cited papers, one by one.

Response to referee:

Thank you for your constructive advice.

Ref. 1 (*Science*, 1997, **276**, 233): Whitesides et al reported that floating objects could self-assemble into regular arrays by adjusting the shapes of the assembling objects and the wettability of their surfaces;

Ref. 2 (*Nat. Nanotech.*, 2017, **12**, 73): Brugger et al harnessed the capillary assembly of Au nanorods to attain simultaneous control of position, orientation and interparticle distance on structured substrates with trap array;

Ref. 3 (*Angew. Chem. Int. Ed.*, 2019, **58**, 5246): Schönherr et al showed an interesting approach that enabled generation of cell microenvironment by capillary assembly of microobjects at an air/water interface;

Ref. 4 (*J. Am. Chem. Soc.*, 2012, **134**, 5801): DeSimone et al demonstrated that Capillary-driven self-assembly of microrods at a water/oil interface could induce the formation of bilayer or ribbon-like structures;

Ref. 5 (*Annu. Rev. Condens. Matter. Phys.*, 2018, **9**, 283): Stebe et al reviewed recent findings in capillary assembly of colloids, and the roles of contact line pinning, particle shape, and surface chemistry in structure formation;

Ref. 6 (*Science*, 2002, **295**, 2418): Whitesides et al proposed that self-assembly is a widely applied strategy in synthesis and fabrication at different length scales. The studies of self-assembly driven by capillary interactions would help us to understand many important structures in biology.

Ref. 7 (*Am. J. Phys.*, 2005, **73**, 817): Mahadevan et al expounded the physical mechanism in the “Cheerios effect” (the phenomenon of attraction or repulsion between floating objects). and presented a simple dynamical model of capillary interaction between floating objects;

Ref. 8 (*Phys. Rev. Lett.*, 2019, **123**, 254502): Harris et al developed a novel method to directly measure the capillary attraction force in the “Cheerios effect”. This work not only provides an effective approach to quantify the capillary attraction, but also deepens the understanding of the “Cheerios effect”.

Among the eight cited papers, Ref 1-6 reported fabrication approaches using self-assembly of floating objects to generate various aggregated structures. These methods all employ capillary action between floating objects to gain ordered aggregation. The six papers suggest that self-assembly of floating objects is an effective approach to prepare ordered structures. Ref 7 revealed the physics in cheerio’s effect. Ref 8 developed a method to measure the capillary force

between floating objects. Ref 6-8 suggested that discovering the physics in capillary action between floating objects (the physics in cheerio's effect) would have important meanings for understanding the formation of aggregation structures on liquid interface in biological systems.

Therefore, we relabeled references in the manuscript to precisely signify that Refs1-6 are significant for bottom-up fabrication of novel materials and devices, while Refs 6-8 suggest that discovery of the mechanism of the assembly on interfaces would help to improve fundamental understanding of biological systems.

Comments 5:

In line 69, the authors start introducing the material system they are using (azo-LCP). Line 71, what means 'splay'?

Response to referee:

Thank you for your question. The term "splayed alignment" means a tilted orientation of LC mesogens as shown in Fig. R4, wherein the nematic director of LC tilts and varies from 0° at the bottom to 90° at the top^[3]. We have added the figure in the revised Supplementary Information in Section 2.

Figure R4. Schematics showing splayed alignment in the LCN. The dashed rectangle indicates the xz plane of the LCN. The inset shows the splayed alignment of LC mesogens: the nematic director of LC mesogens tilts and varies in the xz plane of the LCN, the tilt angle gradually changes from 0° at the bottom to 90° at the top.

[3] White, T. J. & Broer, D. J. Programmable and adaptive mechanics with liquid crystal polymer networks and elastomers. *Nat. Mater.* **14**, 1087-1098 (2015).

Comments 6:

Line 74, lists 23-32 papers on photo-induced changes of this polymer: This number is high and is not justified.

Response to referee:

We appreciate your comment. Ref 23 is a milestone in the research of photodeformation of azobenzene liquid crystal networks (azo-LCNs), in which, Ikeda et al, for the first time, achieved three-dimensional photodeformation of LCN materials. Ref 24 reported a photoaligned method to prepare azo-LCNs with splayed alignment, which show monodirectional bending behavior upon light irradiation. Ref 27 exhibits an inkjet printing method used to prepare light-driven artificial cilia from azo-LCN actuators. Ref 28 demonstrates o-fluorinated azobenzene liquid crystal

polymer materials enabling all optical control of shape deformation and shape restoration. In Ref. 28, Dirk Broer and colleagues developed novel azo-LCN material containing the azobenzenes with rapid *cis-to-trans* thermal relaxation times, which allows continuous oscillating wave motion throughout the LCNs. Ref 31 reported kirigami-based light-induced shape-morphing and locomotion of azo-LCNs. Ref 32 reported a strategy of building three-wavelength modulated liquid crystal polymer actuators capable of performing multi-directional movement. Ref 25, 26, 29, and 30 are the critical papers that review the development of photodeformable LCNs in the last ten years. These important papers were cited to support that azobenzene liquid crystal polymer actuators with parallel or splayed alignment have the ability to dynamically and reversibly transform from flat to bending state through asymmetric contraction or expansion arising from photoinduced conformation changes of azobenzenes.

Comments 7:

Also, nowhere in the paper the authors are discussing the physics behind the opto-capillary effect. There is a nice paper [in International Journal of Solids and Structures 128 (2017) 36–49] that discussed in all detail. The authors of this paper should include some basic information on the mechanism of bending and what forces are involved in the assembly process. It can be done by linking to existing information in other papers already published, to assess the quality of the assembly.

Response to referee:

Thank you for your constructive advice. As discussed in the suggested reference (*Int. J. Solids. Struct.*, 2017, **128**, 36) a multiscale model was proposed for predicting the *trans-cis-trans* reorientation-based photodeformation behavior of the azo-LCP. This reference is relevant, but their mechanism is a little different from ours. The opto-capillary effect in our work can be explained by the *trans-cis* isomerization-based photodeformation mechanism shown as follows.

Figure R5. Photodeformation mechanism of azo-LCN. a. Schematics showing the isomerization of azobenzene. Rod-shaped trans-azobenzene with molecular length of 9.0 Å

absorbs UV light and transforms to bent *cis*-azobenzene with molecular length of 5.5 Å. Irradiated by visible light, *cis*-azobenzene backs to *trans*-azobenzene. **b.** Schematics showing light-induced reversible bending of homogeneous azo-LCN. Upon UV irradiation, LC-to-isotropic phase transition occurs only in the surface region of the LCN, which results in an uneven distribution of the molecular alignment through the thickness of the sheet, and the gradient of photo-induced strains through thickness induces the bending motion toward the incident direction of the actinic UV light.

Azobenzene can undergo a reversible photochemical reaction between two isomers (Fig. R5a). Rod-shaped *trans*-azobenzene with molecular length of 9.0 Å absorbs UV light and transforms to bent *cis*-azobenzene with molecular length of 5.5 Å. If irradiated by visible light, *cis*-azobenzene backs to *trans*-azobenzene. Rod-shaped *trans*-azobenzenes can stabilize the ordered phase structure of the LC, while the bent *cis*-azobenzenes tend to destabilize the phase structure [4]. When azobenzenes are incorporated into liquid crystal network (LCN) as photoresponsive mesogens, the photochemical reaction of azobenzene can be used to isothermally trigger reversible LC-to-isotropic phase transition of the LCN [5].

Azobenzene possesses a high extinction coefficient around ~ 360 nm (about $2.0 \times 10^4 \text{ L mol}^{-1} \text{ cm}^{-1}$). High concentration of azobenzene in the LCN causes extensive absorption of the incident photons at the surface region (99% incident photons are absorbed by the surface within a thickness of < 1 μm), which means that LC-to-isotropic phase transition occurs only in the surface region of the LCN [6]. In the homogeneous LCN, LC mesogens are aligned parallel to the long axis of the LCN. Upon UV irradiation, the LC-to-isotropic phase transition occurred in the surface region of the LCN generates an uneven distribution of the molecular alignment through the thickness of the LCN (Fig. R5b), this means the anisotropic contraction is induced only in the surface region of LCN upon the irradiation, and thus the difference of photo-induced strains through the thickness lead to bending response toward the incident direction of the actinic UV light [7].

We have added the above mechanism analyses in section 2 of the revised Supplementary Information. As for capillary forces involved in the assembly process, the elastic deformation of the azo-LCP caused by them is negligible and they are elaborated in section 4 of the revised Supplementary Information (see the response to comments 10 for detail).

[4] Yu, Y. L. & Ikeda, T. Alignment modulation of azobenzene-containing liquid crystal systems by photochemical reactions. *J. Photoch. Photobio. C.* **5**, 247-265 (2004).

[5] Kondo, M., Yu, Y. L. & Ikeda, T. How Does the initial alignment of mesogens affect the photoinduced bending behavior of liquid-crystalline elastomers? *Angew. Chem. Int. Ed.* 2006, **45**, 1378-1382.

[6] Yoshino, T. et al. Three-dimensional photomobility of crosslinked azobenzene liquid-crystalline polymer fibers. *Adv. Mater.* **22**, 1361-1363 (2010).

[7] Ube, T. & Ikeda, T. Photomobile polymer materials with crosslinked liquid-crystalline structures: molecular design, fabrication, and functions. *Angew. Chem. Int. Ed.* **53**, 10290-10299 (2014).

Comments 8:

line 89/90: what justifies the number of 9 actuator and 19 structures?

Response to referee:

Thank you for your question. There is no connection between 9 actuators and 19 structures. The number of the structures formed by 9 actuators is far more than 19. The reason for choosing 9 actuators is that they can form a rich assortment of assembly structures with diverse morphologies, which can give a good demonstration of the ability of morphology control of the optocapillary approach. We have done a simple estimate. Based on the three basic assembly states (end to end, side to side, and T-shaped state shown in Fig. R6a), two, three, and four actuators can be reconfigured into 3, 12, and 57 assemblies, respectively (Fig. R6). Some experimental photographs of corresponding assembled structures are shown in Fig. R7. When the number of the actuators reaches to 9, the number of assembled structures would be up to tens of thousands. Therefore, only some assembled structures are shown in Fig. 2 (Fig. 5b in the revised manuscript).

Figure R6. Schematics showing light-driven reconfigured structures by 2 actuators (a), 3 actuators (b), and 4 actuators (c), respectively. Interestingly, 6 structures formed by 2 actuators and 40 structures formed by 3 actuators show chirality. The two structures at the left and right sides of the dotted lines are enantiomers that constitute mirror images which cannot be superimposed on each other.

Figure R7. Photographs showing light-driven reconfigured structures by 2 actuators (a), 3 actuators (b), and 4 actuators (c), respectively.

Comments 9:

line 102: what means this sentence with "... cutting-edge research fields"

Response to referee:

Thank you for your question. As optocapillarity-driven reconfiguration exhibits its capability to active control morphologies or patterns of the assembly (Fig. 5 in manuscript), we highlighted such photocontrol of morphologies, which shows potential to find use in the areas demanding for dynamic morphology control. To better present our idea, we have rephrased the sentence to be

“These assemblies and their reconfigurable capability are of great significance for potential applications in the fabrication of structures and devices with tunable mechanical, optical or electronic properties in the fields that demand for dynamic and programmable control of structural topology, such as robotic swarms, advanced sensor devices, and synthetic swimmers.” The related texts in the manuscript have been updated accordingly.

Comments 10:

line 105 onward: the explanation of the mechanism is very basic, and more physics should be included to better understand and predict the system

Response to referee:

Thank you for your constructive advice. We have added more detailed dynamic analysis to explain the assembly and disassembly process of the system in the revised manuscript.

In our experiments, the deformation of the actuator is mainly regulated by the light field. To facilitate the understanding, the actuators can be regarded as rigid bodies during their assembly process. As shown in Fig. R8a, the forces acting on a single floating actuator can be divided into three parts, which are the gravity mg , the hydrostatic pressure ρgz acting on the lower surface of the actuator, and the surface tension γ acting on the three-phase line (TPL), i.e. the boundary of the actuator. Then, the resultant force on the actuator along the vertical direction and the horizontal direction are respectively

$$\vec{F} = m\vec{g} + \int_{TPL} \vec{\gamma} dl + \iint_{WS} \rho gz \vec{n} ds \quad (r3)$$

$$\vec{M} = \int_{TPL} \vec{L}_p \times \vec{\gamma} dl + \iint_{WS} \rho gz \vec{L}_p \times \vec{n} ds \quad (r4)$$

Here \vec{n} is the local normal vector of the actuator surface, \vec{L}_p is the distance vector from the center of gravity to any point on the TPL or actuator surface. Once the actuator is in equilibrium, the adjacent air-water interface has symmetric geometry (Fig. 3a in the revised Manuscript). When another actuator approaches, as shown in Fig. R8b, the morphology of air-water interface changes and the symmetry is broken. Therefore, the direction of surface tension acting on the three-phase line varies accordingly, resulting in capillary force F_c and moment M_c . Noted that the contribution of hydrostatic pressure to the horizontal direction and water resistance could be negligible^[1, 2], hence the horizontal motion of the actuator will be dominated by F_c and M_c . The force and moment can be calculated as the partial derivative of energy (Fig. 2b-2e in the revised Manuscript) with respect to the displacement and the orientation angle, respectively. As shown in Fig. R8c, during the end-to-end assembly, the capillary force between two floating actuators is inversely proportional to their distance and possesses a nonlinear characteristic. For two actuators with the same curvature ($c_1 = c_2$), the greater the bending curvature, the stronger the capillary attraction force between them. However, when one of them is treated with visible light and becomes flat ($c_1 > 0, c_2 = 0$), the capillary force will drop to less than zero. There will be a repulsive force between them, resulting in the disassembly of the end-to-end state. Furthermore, when this actuator bends in the opposite direction, the capillary repulsion will get stronger to speed up the disassembly process. Fig. R8d shows the capillary moment curve of one actuator with respect to the orientation angle when it is near to another one. It illustrates that the two actuators tend to align parallelly when they are bent in the same direction but vertically when they are not.

We have added the above theoretical analyses and discussions in Section 3 of the revised Supplementary Information.

Figure R8. Dynamic analysis of the floating actuators. **a.** Force analysis of the single floating actuator. **b.** Capillary interaction between two floating actuators. **c.** Numerical data for capillary force F_c on the actuator (with curvature c_2) vs relative distance d/l from another one (c_1). Here the long axes of the two actuators are in a line, as shown in the inset. **d.** Numerical data for capillary moment M_c on the actuator (c_2) vs the orientation angle α with another one (c_1). Here the center of the c_2 actuator is on the long axis of the c_1 actuator and the center-to-center distance is $d = 1.2l$, as shown in the inset.

Comments 11:

line 117: typo 'low'

Response to referee:

Thank you for your correction. The “low” has been changed to “lower”.

Comments 12:

line 129 onward: the mechanism of top versus bottom illumination is not well described; maybe a sketch with a drawing and explaining the physico-chemical mechanism would be helpful

Response to referee:

Thank you for your constructive advice. It's known that Azobenzene can undergo a reversible photochemical reaction between two isomers (Fig. R9a). Rod-shaped *trans*-azobenzene with molecular length of 9.0 Å absorbs UV light and transforms to bent *cis*-azobenzene with molecular length of 5.5 Å. Irradiated by visible light, *cis*-azobenzene can back to *trans*-azobenzene. Rod-shaped *trans*-azobenzenes can stabilize the ordered phase structure of the LC, while the bent *cis*-azobenzenes tend to destabilize the phase structure^[4]. When azobenzenes are incorporated into liquid crystal network (LCN) as photoresponsive mesogens, the photochemical reaction of azobenzene can be used to isothermally trigger reversible LC-to-isotropic phase transition of the LCN^[5]. Azobenzene has pretty high extinction coefficient around ~ 360 nm (about 2.0×10^4 L mol⁻¹ cm⁻¹). High concentration of azobenzene in the LCN film causes extensive absorption of the incident photons at the surface region (99% incident photons are absorbed by the surface within a thickness of < 1 μm), which means that LC-to-isotropic phase transition occurs only in the surface region of the LCN upon UV irradiation^[6].

In the homogeneous LCN, LC mesogens are aligned parallel to the long axis of the LCN. As

shown in Fig. R9b, when the top surface of LCN is illuminated by UV light, the LC-to-isotropic phase transition at the top surface causes the anisotropic contraction along the long axis of the LCN, which generate the strain mismatch between the top layer penetrated by UV and deeper layer that UV can't reach. This strain mismatch leads to upward bending of LCN toward the incident direction of the actinic UV light. In the same way, when its bottom surface is illuminated by UV light, the strain mismatch makes the LCN bending downwards.

We have added the mechanism analyses and the sketch in section 2 of the revised Supplementary Information.

Figure R9. Schematics showing photodeformation mechanism of the LCN. **a.** Schematics showing the isomerization of azobenzene. Rod-shaped *trans*-azobenzene with molecular length of 9.0 Å absorbs UV light and transforms to bent *cis*-azobenzene with molecular length of 5.5 Å. Irradiated by visible light, *cis*-azobenzene can back to *trans*-azobenzene. **b.** Schematics showing light-induced directional bending of homogeneous LCN. when its bottom surface is illuminated by UV light, the LC-to-isotropic phase transition of the bottom surface causes the anisotropic contraction along the long axis of the LCN, which generate the strain mismatch between the top layer penetrated by UV and deeper layer that UV can't reach. This strain mismatch leads to downward bending of LCN toward the incident direction of the actinic UV light. In the same way, when the top surface of LCN film is illuminated by UV light, the strain mismatch between the UV-penetrated layer and the deeper layer that UV can't reach makes the LCN bending upwards.

Comments 13:

line 140: finally some numbers on the bending radius as function of the light source intensity; but what is the rational behind? what is the model and physics? it is just an observation but no explanation

Response to referee:

Thanks for your constructive comments. The bending of azo-functionalized liquid crystal polymer actuators (LCN films) is caused by the light-induced uneven shrinkage deformation along the thickness direction (Fig. R10). The relationship between the bending radius and the light intensity have been explained in previous reports^[8-10].

When the film is illuminated by UV light from the top, due to the strong capacity of photo absorption of the material containing azobenzene groups, the light intensity would decay exponentially along the thickness direction, that is^[8]

$$I = I_0 e^{-[z+(t/2)]/d} \quad (r5)$$

Here I_0 is the light intensity at the top surface, z is the distance from the top surface, t is the thickness of the film, and d is the attenuation coefficient. Upon UV irradiation, LC-to-isotropic phase transition occurs, generating negative photostrain within the film. The photostrain is determined by the light intensity I and the light duration T ^[8-10]. Assuming the photostrain is proportional to the light intensity, we have

$$\varepsilon^p = \varepsilon_0^p(I_0, T) e^{-[z+(t/2)]/d} \quad (r6)$$

Here $\varepsilon_0^p(I_0, T)$ is the in-plane photostrain at the top surface. According to ref. 8, the deformation induced by the above photostrain gradient can be approximately calculated as a bilayer, which is composed of a surface layer with constant photostrain and a deeper layer without photostrain, as shown in Fig R10b. The thickness and the value of photostrain of the surface layer can be respectively expressed as

$$t_s = 2d - \frac{t}{e^{t/d} - 1} \quad (r7)$$

$$\varepsilon_s = \varepsilon_0^p(I_0, T) \frac{(e^{t/d} - 1)^2}{2e^{t/d} [(e^{t/d} - 1)d - t]} \quad (r8)$$

In the absence of external forces, the film will form a cylindrical shape with curvature^[8].

$$c = \frac{6\varepsilon_s t_s (t_s - t)}{\pi t^3} = \frac{3(2de^{t/d} - 2d - t)(2de^{t/d} - 2d - te^{t/d})}{\pi t^3 e^{t/d} (de^{t/d} - d - t)} \varepsilon_0^p(I_0, T) \quad (r9)$$

Therefore, the curvature of the film will depend on the light intensity and light duration.

We have added three new references and the above analysis in section 2 of the revised Supplementary Information.

Figure R10. Schematics showing the photodeformation model of azo-functionalized LCN. a. light-induced reversible bending of homogeneous azo-LCN. Upon UV irradiation, LC-to-isotropic phase transition occur only in the surface region of the LCN, which result in an uneven distribution of the molecular alignment through the thickness of the sheet, and the gradient of photo-induced strains through thickness induce the bending motion toward the incident direction of the actinic UV light. **b.** Schematics showing light-induced bending model of the LCN, which can be regarded as a bilayer: a surface layer with constant photostrain and a deeper layer without photostrain.

[8] Dunn, M. L. Photomechanics of mono- and polydomain liquid crystal elastomer films. *J. Appl. Phys.*, **102**, 013506 (2007).

[9] Yun, J. H., Li, C., Chung, H., Choi, J. & Cho, M. Multiscale modeling and its validation of the trans-cis-trans reorientation-based photodeformation in azobenzene-doped liquid crystal polymer. *Int. J. Solids. Struct.*, **128**, 36-49 (2017).

[10] Ikeda, T., Nakano, M., Yu, Y., Tsutsumi, O., & Kanazawa, A. Anisotropic bending and unbending behavior of azobenzene liquid-crystalline gels by light exposure. *Adv. Mater.*, **15**, 201-205 (2003).

Comments 14:

line 207: after assembly, what is the force that keeps them together (and what force is needed to break it apart?)

Response to referee:

Thank you for your important comments. As shown in Fig. R11, the capillary force arising from surface tension acting on the three-phase line generates the attraction (or repulsion) to keep them together (or apart) (see the response to comments 10 for detailed force analysis). The magnitude of the capillary attraction is proportional to the bending curvature of actuators (Fig. R11c). The greater the bending curvature, the stronger the capillary attraction. After assembly, the capillary

attraction always exists among bent actuators when their shapes retain unchanged. Even when their assembly was broken apart by external force (Fig. 4a in the manuscript), the bent actuators allow spontaneous reassembly after removing the external force. After one of the bent actuators returns to the flat state or bend in the opposite direction treated by the light, the capillary force change direction and the capillary repulsion exists, resulting in the disassembly of actuators. Similarly, as shown in Fig. R11d, when there is a minor rotation between the two actuators assembled end to end, it will give rise to a resisting moment leading to a spontaneous recovery. But if one actuator becomes flat or bending in the opposite direction, the moment will instead turn out to accelerate the rotation until they are perpendicular to each other.

We have added the above theoretical analyses and discussions in Section 3 of the revised Supplementary Information.

Figure R11. Dynamic analysis of the floating actuators. a. Force analysis of the single floating actuator. b. Capillary interaction between two floating actuators. c. Numerical data for capillary force F_c on the actuator (with curvature c_2) vs relative distance d/l from another one (c_1). Here the long axes of the two actuators are in a line, as shown in the inset. d. Numerical data for capillary moment M_c on the actuator (c_2) vs the orientation angle α with another one (c_1). Here the center of the c_2 actuator is on the long axis of the c_1 actuator and the center-to-center distance is $d = 1.2l$, as shown in the inset.

Comments 15:

Videos: they are in general very nice and informative; but maybe there are too many; also it seems that for some assembly task the light source has to be adjusted very close and precise, so one wonders that the assembly could also be done by a tweezer tool; sometimes there is strong light coming in to the camera which is not ideal.

Response to referee:

Thank you for your valuable advice. Although the tweezer can be used to adjust the positions and directions of the actuators for a short time, the final stable structure of the system still depends on the capillary forces induced by the bending of the actuators. Besides, when a tweezer touches the water surface, the menisci surrounding the tweezer attracts or repel the menisci of actuators. As a result, the actuators were either adhered to the tweezer or repelled away, which lead to the failure in assembly of the actuators (Supplementary Movie R1). Therefore, it is invalid to execute the programmable assembly and reconfiguration using a tweezer.

Moreover, we have optimized the distance between the light source and actuators, and updated the videos and the photographs (Fig. R12), in which the visual disturbance by the dazzling light has been already eliminated (Supplementary Movie 8). And the Figure in the manuscript is updated accordingly. In addition, we have merged some videos, and the total number of supplementary videos were reduced from 12 to 10.

Figure R12 | Optocapillarity-driven programmable reconfiguration. Photographs showing light-induced assembly and reconfiguration of nine rectangular actuators into diverse structures. Upon light irradiation, the actuators assemble into regular structures. The transformation between any two structures is achieved by alternative localized irradiation of 450-nm and 360-nm laser as well as modulation of the incident direction of the laser (Supplementary Movie 8). The inset at the top right corner of each photograph schematically indicates the assembled structure. The intensity of the ultraviolet laser and visible laser is 60 mW cm^{-2} and 45 mW cm^{-2} , respectively. The size of

rectangular actuators is $6 \text{ mm} \times 2 \text{ mm} \times 0.03 \text{ mm}$.

Comments 16:

video 4 shows some bubbles; what are they?

Response to referee:

Thank you for your valuable observation. We have checked the setup of the experiment and found that it's an air bubble in the bottom wall of the glass container (Fig. R13). The bubble may be generated in the wall during production of the glass container.

Figure R13. Photograph showing the air bubble in the bottom wall of the glass container.

Comments 17:

figure 1: ial: later view should be 'lateral view'

Response to referee:

Thank you for your correction. The “later view” was already corrected to 'lateral view'.

Comments 18:

figure 2: why these numbers 9 and 19?

Response to referee:

Thanks for your professional comments. Please see the response of comment 8.

Comments 19:

The paper currently lacks the physics explanation of the effect, without which it is not yet ready for publication.

Response to referee:

Thanks for your professional advice. In the revised manuscript, as elaborated in the responses to previous comments, we have added a new section to clarify the underlying physical mechanisms of the light-induced deformation, and more detailed dynamic analysis of single and two floating actuators to explain the opto-capillary assembly and disassembly process. We hope the reviewer will find the physical mechanisms explained more clearly in the revised manuscript.

Response to referee 2

Comments 1:

The paper by Hu et al. "Optocapillarity-driven assembly and reconfiguration of liquid crystal polymer actuators" shows an interesting application of liquid crystal elastomers at the water interface. The elastomer stripes, which can be bent by light due to the presence of azobenzene, change their shape and thus change their capillary interactions. With this stratagem the authors obtain a tunable "cheerios effect" and are able to create reversible assemblies of elastomer stripes (and other shapes) at the interface by regulating the wavelength and the direction of the illumination. I enjoyed reading this paper very much, and I think this research is interesting and worthwhile. However, I cannot recommend it for publication on Nature Communications for the following reasons.

(1) Despite the fact that these experiments have not been done before, to the best of my knowledge, I cannot find anything novel enough in the paper to grant its publication on Nature Communications. Reversible assemblies of soft materials at interface was shown in a system with a richer behavior by Bae et al. Materials Horizons 2017 (ref. 10 in the manuscript). The bending of azobenzene-containing elastomers is very well controllable and well known (I realize this is not the main point of the paper). Most importantly, the capillary interactions between objects with the shape of the elastomers have also been extensively characterized (Loudet et al, PRL 2005, Lewandovsky et al. Soft Matter 2009, Hu & Bush, Nature 2005 and others). This paper combines these elements, but I think the novelty is limited and therefore I would recommend the paper for publication in a more materials-focused journal.

Response to referee:

We are pleased that the reviewer found our work interesting. We are also grateful for his/her statement that the reported experiments had not been demonstrated before in literature. We agree with the reviewer that reversible assemblies using capillary interactions have been demonstrated previously. However, as elaborated below, our work **highlights an optocapillary approach that actively uses "cheerios effect" to realize tunable and reversible reconfigurations of floating objects and enable active manipulation of individual component to form diverse assemblies with adjustable topological structures**. We kindly request the reviewer to reconsider the unique features of our work in a broader context as outlined below:

(1) **Active photocontrol of topological structures of assemblies through photo-driven directional bending of liquid crystal networks (LCNs) actuators by regulation of actinic light (direction, intensity, location)**. Whereas in previous reports, assembly structures are determined by inherent shapes of floating objects. Once the assembling is finished, the obtained assembly structure is fixed owing to their constant shapes (*Phys. Rev. Lett.*, 2005, **94**, 018301; *Soft Matter*, 2009, **5**, 886). Similarly, in the work of Bae et al (*Mater. Horiz.*, 2017, **4**, 228), the assembling structure or pattern is defined once by the prescribed shapes of hydrogel sheets via the swelling pattern formed by grayscale lithography in advance;

(2) **Construction of numerous assembly structures by local/modular manipulation**. In our system, assembly structures are analogous to a pattern composed of many pixels. Each pixel is made of two adjacent actuators, and its state is determined by the assembly state of the two

adjacent actuators (end-to-end, vertical, and side-by-side) (Fig. R14). Adjusting assembly state of each pixel, we can control the topological shape of the entire pattern. By increasing the number of pixels, we can gain a wide variety of assembly patterns with different topological shapes. Fig. R15 shows that two, three, and four actuators can build 3, 12, and 57 structures, respectively. Some representative structures are shown in Fig. R16. Such optocapillarity approach exhibits a high potential for generating diverse building blocks with rich morphologies. For example, as shown in Fig. R17, four actuators can be light-driven to be reversibly reconfigured into L and Z-shaped assemblies, like tetrominoes of the popular video game Tetris. Moreover, our optocapillarity approach also shows the capability to construct a rich assortment of chiral structures (Fig. R15b and R15c), and even actively tune the chirality of assembly (Fig. R18), which offers a versatile platform capable of bottom up engineering of switchable chiral metamaterials.

(3) **Light response of our system allows ease of contactless, spatial, localized, temporal, isothermal control of assembling structures.** As show in Fig. 4 in the manuscript, the optocapillarity approach can control individual objects within a large ensemble, while this kind of control is difficult to achieve in thermal-responsive approaches (*Mater. Horiz.*, 2017, **4**, 228). Moreover, the assembly behaviors in our system are controlled through the deformation mechanism based on the isothermal photochemical phase transition of azobenzene-containing LCNs, and the isothermal response is crucial for extending an approach to apply in biomedical and bioengineering applications.

Figure R14. Optocapillarity-driven dynamic reconfiguration among three assembly state: end-to-end assembly, T-shaped assembly and side-by-side assembly. In each state enclosed by the dashed rectangle, the top images are the schematics showing the assembly states of the two actuators, while and the below images are the corresponding experimental photographs.

Figure R15. Schematics showing light-driven reconfigured structures by 2 actuators (a), 3 actuators (b), and 4 actuators (c), respectively. Interestingly, 6 structures formed by 2 actuators and 40 structures formed by 3 actuators show chirality. The two structures at the left and right sides of the dotted lines are enantiomers that constitute mirror images that cannot be superimposed on each

other.

Figure R16. Photographs showing light-driven reconfigured structures by 2 actuators (a), 3 actuators (b), and 4 actuators (c), respectively.

Figure R17. Photocontrol of four rectangular actuators to assemble into structures like the tetriminoes of the popular video game Tetris. The L and Z-shaped structures have chirality which can be switched from one enantiomer to another by light-driven reconfiguration. The first

image shows the tetrominoes of the video game Tetris.

Figure R18. Photographs showing light-driven switching of the chirality of assembled structures. 4 actuators formed two types of key-shaped assemblies (a and b). Each type of the assemblies exists two enantiomers which can be switched by light-driven reconfiguration of the actuators.

Comments 2:

(2) One of the main claims of the paper is how well controllable this system is. The videos however do not fully support this claim, in my opinion. For example, upon irradiation elastomers seems to fall either in state 1 or 3 almost randomly (side-to-side or head-to-tail assembly). The energy difference between these two states seen in simulations is not supported by a study of how frequently the two modes of assembly are observed. One of the elements which I cannot find in the paper is the importance of the initial relative position of the two elastomers when they are irradiated. This was definitely important in the assembly of anisotropic particles and I expect it to be a relevant parameter also here.

Response to referee:

Thank you for your valuable comments. We are sorry that we did not explain the mechanism as clearly as possible. The assembly process is controlled by the bending of actuators. When the bending direction of two actuators is opposite, one bends upwards and the other bends downwards, they always adopt T-shaped assembly. When two actuators have the same bending direction, both bend upwards or downwards. The resulted state of assembly will be affected by initial distance between them, as speculated by the reviewer. There is a critical distance that determines assembly states. As shown in Fig. R19, Under the premise of radiating light with the same intensity (110 mW cm^{-2}) and placing two actuators in parallel, when the initial distance between two actuators is greater than the critical distance $\sim 11.2 \text{ mm}$, two actuators form end to end assembly, while the distance is less than the critical distance and they would like to aggregate into side-by-side.

Although the assembly may be affected by the initial positions of the actuators, the configuration can be manipulated under well control. As shown in Fig. R20, to achieve the reconfiguration from end-to-end assembly (a1) to T-shaped assembly (a3), one actuator was illuminated with visible light to turn to flat (a2, b2), and then its bottom surface was irradiated with UV light, and the actuator bent downwards, then the two actuators aggregate into T-shaped assembly (a3, b3). For the reversible reconfiguration from T-shaped assembly (a3) to end-to-end assembly (a5), we can

firstly illuminate downward bending actuator with visible light to make it flat, and then the flattened actuator rotated (a4, b4) owing to the capillary-induced alignment. When the angles of long axis of the two actuators was less than 60° , the flattened actuator was illuminated UV light to bend upwards, and then the two actuators reaggregated back into end-to-end assembly (a5, b5). To achieve the reconfiguration from end-to-end assembly to side by side assembly, firstly we can illuminate one actuator with visible light to make it flat, and the flattened actuator rotated (a6, b6). When the orientation angle of the two actuators was large than 120° , the flattened actuator was illuminated with UV light to let it bend upwards, and then the two actuators formed side by side assembly (a7, b7). To gai the reconfiguration from side by side assembly to T-shaped assembly, one actuator was irradiated with visible light to become flat (a8, b8). The flattened actuator was irradiated with UV light at its bottom surface to make it bend downwards. Then the two actuators aggregated into T-shaped assembly (a9, b9). Using the same methods described above, the T-shaped assembly can be reconfigured again into end-to-end assembly (a9-a10, b9-b10). The operator can precisely produce the desired assembly structure through the light-driven modulation of directional bending of the actuators.

We have added the abovementioned discussion in section 3 of the revised Supplementary Information.

Figure R19. Diagram showing the assembly states determined by the initial distance between two actuators. Schematics below the diagram showing the states of two actuators before and after the assembly. d_c is the critical distance that determines assembly states. When the initial distance (d) between two actuators is greater than d_c , two actuators form end to end assembly, while the distance (d) is equal to or less than d_c , they would like to form side-by-side assembly.

Figure R20. Experimental photographs (a) and corresponding Schematics (b) showing optocapillarity-driven reconfiguration of two actuators. To achieve the reconfiguration from end-to-end assembly (a1) to T-shaped assembly (a3), one actuator was illuminated with visible light to make it flat (a2, b2), and then its bottom surface was irradiated with UV light (a3, b3), and the actuator bent downwards, then the two actuators aggregated into T-shaped assembly. For the reversible reconfiguration, we can firstly illuminate downward bending actuator with visible light to recover to flat (a4, b4), and then the flattened actuator rotates (a4, b4). When the angles of long axis of the two actuators was less than 60° , the flattened actuator was illuminated UV light to bend upwards, and then the two actuators reaggregated back into end-to-end assembly (a5). To achieve the reconfiguration from end-to-end assembly to side by side assembly, firstly we can illuminate one actuator with visible light to make it flat, and the flattened actuator rotated (a6, b6). When the orientation angle was large than 120° , the flattened actuator was illuminated with UV light to let it bend upwards, and then the two actuators formed side by side assembly (a7, b7). To gain the reconfiguration from side by side assembly to T-shaped assembly, one actuator was irradiated with visible light to become flat (a8, b8). The flattened actuator was irradiated with UV light at the bottom surface to make it bend downwards. Then the two actuators aggregated into T-shaped assembly (a9, b9). Using the same methods described above, the T-shaped assembly can be reconfigured again into end-to-end assembly (a9-a10, b9-b10).

Comments 3:

The 19 configurations shown in figure 2, likewise, seem quite arbitrary, and from the video they seem to be produced by a series of trials and errors in the assembly, but there is no analysis of this in the paper (for examples, which switches are very favorable and reliable, and which are not?).

Response to referee:

Thank you for your comments. The reconfiguration of aggregated structures of multiple actuators is based on the switching among three primary assembly states (end to end, T-shaped, and side by side assemblies). As shown in the Fig. R21, the three assembly states can be reversibly reconfigured to each other by light-driven directional bending of the two actuators. When we use multiple actuators, we can dynamically tune the overall morphology of the assembly through

adjusting the assembly states of each two adjacent actuators. During structure transitions of multiply actuators, the system has multiple steady states with similar energies after the curvature changes occur in one or more actuators. And the system has multiple transition routes, which, however, is not an error. The greater the number of actuators in the system, the more likely these multiple stable states will appear. Although such multiple stable states will bring limited randomness to the transition processes, but through feedback adjustment, we can successfully achieve the final target structure (Supplementary Movie 8).

Moreover, similarly like the Transformers require several intermediate steps to adjust their shape to reach the final Auto shape. When using many more actuators, we need to adjust the assembly states between adjacent actuators step-by-step to achieve the change of the overall aggregation structures. As shown in Fig. R22 (Supplementary Movie R2), when we aim to gain the reconfiguration of aggregation structure from 1 to 4. We need undergo two steps, the first step to get the structure 2, and then the second step to gain structure 3, and finally produce structure 4. During this process, all steps are favorable and reliable, and can be repeated and reversely operated. It's worth to note that the step 2 and step 3 can be operated simultaneously to reduce the assembly time. In the later research, we plane to use machine vision to recognize the actuators, and patterned lights programmed controlled by computer would enable simultaneous reconfiguration of assembly states among many actuator couples, which would greatly improve the efficiency of the morphological transformation.

Figure R21. Optocapillarity-driven dynamic reconfiguration among three assembly state: end-to-end assembly, T-shaped assembly and side-by-side assembly. In each state enclosed by the dashed rectangle, the top images are the schematics showing the assembly states of the two actuators, while and the below images are the corresponding experimental photographs.

Figure R22. Time-sequence photographs showing light-driven reconfiguration of assembly of nine actuators from structure 1 to structure 4 (Supplementary Movie R2).

Comments 4:

The supplemental material contains information on the curvature of the elastomer as a function of irradiation time, but an example of how the difference in mean curvature (with the same sign) affects the assembly is missing.

Response to referee:

Thank you for your valuable comment. The bending curvature of the actuators plays important role on the strength of the assembly between two actuators. The greater the bending curvature, the stronger the assembly. As shown in the Fig. R23 (Supplementary Movie R3), two bent actuators kept head-to-tail assembly even subjected to the shearing force induced by stirring. But when the actuators were irradiated by visible light, their bending curvature decrease, and the capillary attraction between the two actuators was diminished. As a result, the shearing force can break the head-to-tail assembly, and make it transition to the side to side assembly. The experiment indicates the strength of the assembly can be tuned by light-induced bending curvature of the actuators. Moreover, the bending curvature also affects the response time of the assembly. The greater the bending curvature, the shorter the time required to complete the aggregation. As shown in the Fig. R24, under the premise of maintaining the spacing of 15 mm between two actuators, the time for completing aggregation reduced from 9 s to 4 s as the bending curvature become larger through the increasing the intensities of light irradiation from 90 to 170 mW cm⁻².

Furthermore, we have added a detailed force analysis to explain the above experimental phenomena. In our experiments, the deformation of the actuator is mainly regulated by the light field. To facilitate the understanding, the actuators can be regarded as rigid bodies during their assembly process. As shown in Fig. R25a, the forces acting on a single floating actuator can be divided into three parts, which are the gravity mg , the hydrostatic pressure ρgz acting on the lower surface of the actuator, and the surface tension γ acting on the three-phase line (TPL), i.e. the boundary of the actuator. Then, the resultant force on the actuator along the vertical direction and the horizontal direction are respectively

$$\vec{F} = m\vec{g} + \int_{TPL} \vec{\gamma} dl + \iint_{WS} \rho gz \vec{n} ds \quad (r10)$$

$$\vec{M} = \int_{TPL} \vec{L}_p \times \vec{\gamma} dl + \iint_{WS} \rho gz \vec{L}_p \times \vec{n} ds \quad (r11)$$

Here \vec{n} is the local normal vector of the actuator surface, \vec{L}_p is the distance vector from the center of gravity to any point on the TPL or actuator surface. Once the actuator is in equilibrium, the adjacent air-water interface has symmetric geometry (Fig. 3a in the revised Manuscript). When another actuator approaches, as shown in Fig. R25b, the morphology of air-water interface changes and the symmetry is broken. Therefore, the direction of surface tension acting on the three-phase line varies accordingly, resulting in capillary force F_c and moment M_c . Noted that the contribution of hydrostatic pressure to the horizontal direction and water resistance could be negligible^[11, 12], hence the horizontal motion of the actuator will be dominated by F_c and M_c . The force and moment can be calculated as the partial derivative of energy (Fig. 2b-e in the revised Manuscript) with respect to the displacement and the orientation angle, respectively. The numerical results are shown in Fig. R25c and R25d, for two actuators with the same curvature ($c_1 = c_2$), the greater the bending curvature, the stronger the capillary force or moment between them. Therefore, the capillary attraction force can be increased to speed up the assembly process by increasing the curvature of two actuators.

We have added the above discussions and theoretical analyses in Section 3 and 4 of the revised Supplementary Information.

Figure R23. Time-sequence photographs showing light-driven assembly transition. Two actuators aggregated after UV irradiation retained the head-to-tail assembly when the water interface was stirred at the speed of 200 rpm. when illuminated with visible light, the two actuators immediately transformed from the head-to-tail to side-to-side assembly (Supplementary Movie R3). The white bar in the photographs is a magnetic agitator.

Figure R24. Plots showing center-to-center distance between two actuators versus irradiation time when the intensity of UV light source is different. $I_1 = 170 \text{ mW cm}^{-2}$ (blue squares), $I_2 = 155 \text{ mW cm}^{-2}$ (red rhombs), $I_3 = 120 \text{ mW cm}^{-2}$ (green triangles), and $I_4 = 90 \text{ mW cm}^{-2}$ (orange circles). Schematics above showing experiment setup. The initial center-to-center distance of two actuators is fixed at 15 mm, the distance decreases to 6 mm when they finish the end-to-end assembly upon light irradiation. The size of the actuators is $6 \text{ mm} \times 2 \text{ mm} \times 0.03 \text{ mm}$.

Figure R25. Dynamic analysis of the floating actuators. a. Force analysis of the single floating actuator. b. Capillary interaction between two floating actuators. c. Numerical data for capillary force F_c on the actuator (with curvature c_2) vs relative distance d/l from another one (c_1). Here the long axes of the two actuators are in a line, as shown in the inset. d. Numerical data for capillary moment M_c on the actuator (c_2) vs the orientation angle α with another one (c_1). Here the center of the c_2 actuator is on the long axis of the c_1 actuator and the center-to-center distance is $d = 1.2l$, as shown in the inset.

[11] Vella, D. & Mahadevan, L. The “Cheerios effect”. *Am. J. Phys.* 73, 817-825, (2005).

[12] Yu, Y., Guo, M., Li, X. & Zheng, Q.-S. Meniscus-climbing behavior and its minimum free-energy mechanism. *Langmuir* 23, 10546-10550 (2007).

Comments 5:

The observation of the system stability under stirring is extremely interesting, but for example I can see that some head-to-tail assembly becomes side-to-side after stirring, a phenomenon which was not explained.

Response to referee:

Thank you for your valuable comment. There is an energy barrier that separates the energy levels of head-to-tail assembly and side-to-side assembly (Fig. R26). Energy must be provided to start the transition between these two assemblies. This energy, which is recovered as the transition proceeds, is called activation energy (E_a).

Figure R26. Diagram of the free energy of assembly. An energy barrier exists and separates the energy levels of head-to-tail assembly and side-to-side assembly. Energy must be provided to the

head-to-tail assembly to overcome the energy barrier, which is recovered when the side-to-side assembly are formed. E_a is the activation energy.

In our system, the activation energy is proportional to the bending curvature of actuators. The greater the bending curvature, the higher the activation energy between head-to-tail assembly and side-to-side assembly. As shown in Fig. R27a (Supplementary Movie R4), two actuators at an air/water interface were subjected to the shear force induced by stirring. They can maintain the head-to-tail assembly as the shearing force was increasingly enhanced through accelerating the stirring speed from 100 r/min to 300 r/min. When the stirring speed reached 400 r/min, the shear force was strong enough to induce the transition from the head-to-tail to side to side assembly. This experiment suggests that the transition only occurs when the external energy provided by the stirring is strongly enough to overcome the energy barrier. As shown in the Fig. R27b, two bent actuators kept head-to-tail assembly upon stirring at 200 rpm. When the two actuators were irradiated with visible light which induced the decrease in bending curvature of the actuators, the transition from the head-to-tail to side to side assembly occurred. This experiment indicates the activation energy required for the transition can be tuned by light-driven modulation of the bending curvature of the actuators.

Figure R27. photographs showing tunable transition from head-to-tail assembly and side-to-side assembly. a. Time-sequence photographs showing assembly transition induced by increasing shearing force of stirring. Two actuators with the head-to-tail assembly were located at an air/water interface. The stirring through a magnetic agitator generates a shearing force that can't break the assembly when the stirring speed below 400 r/m. The transition from the head-to-tail assembly to side-to-side assembly occurred until the stirring speed reached 400 rpm (Supplementary Movie R4). b. Time-sequence photographs showing light-driven tunable transition from the head-to-tail assembly and side-to-side assembly. Two bending actuators retained the head-to-tail assembly when the water interface was stirred at the speed of 200 rpm. When illuminated with visible light, the two actuators immediately changed from the head-to-tail assembly to side-to-side assembly (Supplementary Movie R3). The white bar in the photographs is a magnetic agitator.

Comments 6:

Given these objections I don't recommend publication on this journal but I would definitely recommend publication on a more specialized journal. However, I thought that the final part about the assembly over multiple layers of liquids, and multiple interfaces, was really novel and inspiring. Maybe the authors could further expand on this part and explore in details the

capability of this system for 3D assembly, and publish it separately.

Response to referee:

We appreciate the comment of the reviewer. We are very encouraged that the reviewer found the assembly over multiple layers of liquids really novel and inspiring. As elaborated in our responses to Comment-1, *our work presents a novel approach that uses “cheerios effect” to realize tunable and reversible manipulation of the topologies of assemblies through light-driven shape-programmed actuators. Active tuning with a capability of local/modular manipulation to achieve diverse configurations is the unique feature of our method.* We hope the reviewer can reconsider our work in the broader context. Furthermore, motivated by the reviewer, we have improved the 3D assembly and achieved programmable 3D assembly with tunable spatial topologies (Fig. R28). It’s anticipated that such optocapillary approach can be developed to serve as a versatile fabrication platform capable of meeting emerging needs in robotic swarms, advanced sensor devices, chiroptical devices, hierarchical construction, and exotic metamaterials, and many more areas that are beyond the reach of current technologies.

Figure R28. Schematics (a) and corresponding photographs (b) showing optocapillarity-driven 3D synergistic assemblies and reconfiguration. Two rectangular actuators

were placed at the air-water interface while a longer rectangular actuator and another two rectangular actuators was placed at the water/FC-70 interface below the air-water interface. Upon light irradiation the five actuators assembled into a 3D hierarchical structure (Supplementary Movie 10). The first image in each row of images exhibits the initial location of the actuators before light irradiation. Upon light irradiation, the actuators aggregated and formed regular structures. The transformation of the structures was achieved by the localized irradiation of 450-nm and 360-nm laser with modulation of the incident direction. The intensity of the ultraviolet laser and visible laser is 60 mW cm^{-2} and 45 mW cm^{-2} , respectively

Response to referee 3

Comments 1:

This contribution from Lv and colleagues titled "Optocapillarity-driven assembly and reconfiguration of liquid crystal polymer actuators" is a distinctive contribution that demonstrates the use of photoinduced control of the shape of rectangular elements composed of azobenzene-functionalized liquid crystalline polymer networks affects the wetting of these materials and is a means to optically control assembly via capillarity. Further, the work includes modeling, that supports the justification of the assembly from a thermodynamics perspective.

Response to referee:

We highly appreciate your professional comments.

Comments 2:

In revision or resubmission elsewhere, the authors should: 1-describe the mechanics of the deformation of the single films more clearly. Although there is considerable literature on the subject, an illustration of the splay geometry and specific discussion of the means by which this enables deformation is important

Response to referee:

Thank you for your constructive advice. Azobenzene can undergo a reversible photochemical reaction between two isomers (Fig. R29a). Rod-shaped *trans*-azobenzene with molecular length of 9.0 Å absorbs UV light and transforms to bent *cis*-azobenzene with molecular length of 5.5 Å. Irradiated by visible light, *cis*-azobenzene can back to *trans*-azobenzene. Rod-shaped *trans*-azobenzenes can stabilize the ordered phase structure of the LC, while the bent *cis*-azobenzenes tend to destabilize the phase structure^[13]. When azobenzenes are incorporated into liquid crystal network (LCN) as photoresponsive mesogens, the photochemical reaction of azobenzene can be used to isothermally trigger reversible LC-to-isotropic phase transition of the LCN^[14]. Azobenzene possesses a high extinction coefficient around ~ 360nm (about $2.0 \times 10^4 \text{ L mol}^{-1} \text{ cm}^{-1}$). High azobenzene-concentration in the LCN film causes extensive absorption of the incident photons at the surface region (99% incident photons are absorbed by the surface within a thickness of < 1 μm), which means that LC-to-isotropic phase transition occur only in the surface region of the LCN upon UV irradiation^[15].

In the homogeneous LCN, LC mesogens are aligned parallel to the long axis of the LCN (Fig. R29b). When the top surface of LCN is illuminated by UV light, the LC-to-isotropic phase transition of the top surface causes the anisotropic contraction along the long axis of the LCN, which generate the strain mismatch between the top layer penetrated by UV and deeper layer that UV can't reach. This mismatch of photo-induced strain leads to upward bending of LCN toward the incident direction of the actinic UV light. In the same way, when its bottom surface is illuminated by UV light, the mismatch between the UV-penetrated layer and the deeper layer that UV can't reach makes the LCN bending downwards.

Figure R29. Schematics showing photodeformation mechanism of the LCN. **a.** Schematics showing the isomerization of azobenzene. Rod-shaped trans-azobenzene with molecular length of 9.0 Å absorbs UV light and transforms to bent cis-azobenzene with molecular length of 5.5 Å. Irradiated by visible light, cis-azobenzene can back to trans-azobenzene. **b.** Schematics showing light-induced directional bending of homogeneous LCN. when its bottom surface is illuminated by UV light, the LC-to-isotropic phase transition of the bottom surface causes the anisotropic contraction along the long axis of the LCN, which generate the strain mismatch between the top layer penetrated by UV and deeper layer that UV can't reach. This strain mismatch leads to downward bending of LCN toward the incident direction of the actinic UV light. In the same way, when the top surface of LCN film is illuminated by UV light, the strain mismatch between the UV-penetrated layer and the deeper layer that UV can't reach makes the LCN bending upwards.

Figure R30. Schematics showing splayed alignment in the LCN. The small dashed rectangle indicates the xz plane of the LCN. The inset enclosed in an enlarged dashed rectangle shows the splayed alignment of LC mesogens: the nematic director of LC mesogens tilts and varies in the xz plane of the LCN, the tilt angle is gradually changed from 0° at the bottom to 90° at the top.

Figure R31. Schematics showing light-driven bending of the LCN film with splayed alignment. when the top surface of LCN is illuminated by UV light, the LC-to-isotropic phase transition causes the anisotropic expansion along the long axis of the LCN owing to out-plane alignment of LCs at its top surface, which generate the strain mismatch between the top layer penetrated by UV and deeper layer that can't reach by UV. This strain mismatch leads to downward bending of the LCN away the incident direction of the actinic UV light. When its bottom surface is illuminated by UV light, the LC-to-isotropic phase transition generates the anisotropic contraction along the long axis of the LCN owing to in-plane alignment of LCs at the bottom surface of LCN, then the strain mismatch, between the layer penetrated by UV and deeper layer that UV can't reach, results downward bending.

Splayed alignment describes a director profile of LC mesogens aligned in a tilted orientation (Fig. R30), in which the nematic director of LC tilts and varies in the direction perpendicular to the film plane, the tilt angle is gradually changed from 0° at the bottom to 90° at the top.

As shown in Fig. R31, when the top surface of LCN film with splayed alignment is illuminated by UV light, the LC-to-isotropic phase transition of the top surface causes the anisotropic expansion along the long axis of the LCN owing to out-plane alignment of LCs at the top surface of LCN, which generate the strain mismatch between the top layer penetrated by UV and deeper layer that UV can't reach. This mismatch of photo-induced strain leads to downward bending of LCN away the incident direction of the actinic UV light. While its bottom surface is illuminated by UV light, the LC-to-isotropic phase transition of the bottom surface generates the anisotropic contraction along the long axis of the LCN owing to in-plane alignment of LCs at the bottom surface, then the mismatch of photo-induced strain lead to downward bending.

[13] Yu, Y. L. & Ikeda, T. Alignment modulation of azobenzene-containing liquid crystal systems by photochemical reactions. *J. Photoch. Photobio. C.* **5**, 247-265 (2004).

[14] Kondo, M., Yu, Y. L. & Ikeda, T. How Does the initial alignment of mesogens affect the photoinduced bending behavior of liquid-crystalline elastomers? *Angew. Chem. Int. Ed.* **2006**, **45**, 1378-1382.

[15] Yoshino, T. et al. Three-dimensional photomobility of crosslinked azobenzene liquid-crystalline polymer fibers. *Adv. Mater.* **22**, 1361-1363 (2010).

Comments 3:

2-further, the dynamics of the photoinduced response of these elements is an important consideration and should be discussed. The photographs of the process in the SI are insufficient. How do the materials response over long exposures? Is the deformation stable to continuous exposure of light?

Response to referee:

Thank you for your valuable advice. The bending of azo-functionalized liquid crystal polymer actuators (LCN films) is caused by the light-induced uneven shrinkage deformation along the thickness direction. When the film is illuminated by UV light from the top, due to the strong capacity of photo absorption of the material containing azobenzene groups, the light intensity would decay exponentially along the thickness direction, that is ^[16]

$$I = I_0 e^{-[z+(t/2)]/d} \quad (r12)$$

Here I_0 is the light intensity at the top surface, z is the distance from the top surface, t is the thickness of the film, and d is the attenuation coefficient. Upon UV irradiation, LC-to-isotropic phase transition occur to generate negative photostrain within the film. The photostrain is determined by the light intensity I and the light duration T ^[16-18]. Assume the photostrain is proportional to the light intensity, we have

$$\varepsilon^p = \varepsilon_0^p(I_0, T) e^{-[z+(t/2)]/d} \quad (r13)$$

Here $\varepsilon_0^p(I_0, T)$ is the in-plane photostrain at the top surface. According to ref. 16, the deformation induced by the above photostrain gradient can be approximately calculated as a bilayer, which is composed of a surface layer with constant photostrain and a deeper layer without photostrain, as shown in Fig. R32.

Figure R32. Schematics showing the photodeformation model of azo-functionalized LCN. a. light-induced reversible bending of homogeneous azo-LCN. Upon UV irradiation, LC-to-isotropic

phase transition occur only in the surface region of the LCN, which result in an uneven distribution of the molecular alignment through the thickness of the sheet, and the gradient of photo-induced strains through thickness induce the bending motion toward the incident direction of the actinic UV light. b. Schematics showing light-induced bending model of the LCN, which can be regarded as a bilayer: a surface layer with constant photostrain and a deeper layer without photostrain.

The thickness and the value of photostrain of the surface layer can be respectively expressed as

$$t_s = 2d - \frac{t}{e^{t/d} - 1} \quad (r14)$$

$$\varepsilon_s = \varepsilon_0^p(I_0, T) \frac{(e^{t/d} - 1)^2}{2e^{t/d} [(e^{t/d} - 1)d - t]} \quad (r15)$$

In the absence of external forces, the film will form a cylindrical shape with curvature ^[16].

$$c = \frac{6\varepsilon_s t_s (t_s - t)}{\pi t^3} = \frac{3(2de^{t/d} - 2d - t)(2de^{t/d} - 2d - te^{t/d})}{\pi t^3 e^{t/d} (de^{t/d} - d - t)} \varepsilon_0^p(I_0, T) \quad (r16)$$

Therefore, the curvature of the film will depend on the light intensity and light duration. To test deformation stability of the LCN continuously exposed to UV light, we have performed the experiment. As shown in the Fig. R33, Upon UV light irradiation, the LCN reached the maximum bending curvature at 180 s. The LCN maintained a stable bending state when exposed to continuous irradiation.

Figure R33. photographs showing continuous irradiation of UV light at LCN. The intensity of UV light is 120 mW cm^{-2} ; the size of the actuator is $6 \text{ mm} \times 2 \text{ mm} \times 0.03 \text{ mm}$.

[16] Dunn, M. L. Photomechanics of mono-and polydomain liquid crystal elastomer films. *J. Appl. Phys.*, **102**, 013506 (2007).

[17] Yun, J. H., Li, C., Chung, H., Choi, J. & Cho, M. Multiscale modeling and its validation of the trans-cis-trans reorientation-based photodeformation in azobenzene-doped liquid crystal polymer. *Int. J. Solids. Struct.*, **128**, 36-49 (2017).

[18] Ikeda, T., Nakano, M., Yu, Y., Tsutsumi, O., & Kanazawa, A. Anisotropic bending and unbending behavior of azobenzene liquid crystalline gels by light exposure. *Adv. Mater.*, **15**, 201-205 (2003).

REVIEWERS' COMMENTS:

Reviewer #1 (Remarks to the Author):

the authors have answered very extensively to the point raised by this reviewer and have provided additional information that are needed to understand the physics of the assembly. The authors have also spend considerable effort to improve certain paper material including videos. It can now be published, and it will hopefully raise interest in the community to give more feedback.

Reviewer #2 (Remarks to the Author):

I am reviewing the amended version of the manuscript "Optocapillarity-driven assembly and reconfiguration of liquid crystal polymer actuators" by Hu et al.

I would like to reiterate that the experiments are indeed very impressive and that the work is worth publishing. I read from the journal scope that "papers published by the journal aim to represent important advances of significance to specialists within each field". I still claim that the important advance of this paper is the formation of structures across multiple layers. I would like to respond to the authors' answers in detail.

a) The effect shown by the LC elastomers was demonstrated in hydrogels as the authors mention. The objection of the authors is that the prescribed shape of the hydrogel is decided in advance by lithography. I argue that the alignment of the LC elastomer has the same role, as it decides how and in which direction the elastomer will bend with light. The difference is that it can be sometimes bent in two opposite directions.

b) Moreover, the authors emphasize their control on topology and I argue that that is mostly not the case. Almost all the assembly the authors show have the same topology, which is a misrepresentation throughout the text, as I mention also below.

c) I find the "tetris" experiment interesting. However, it is very clear from the video that in order to get to those structure the assembly of multiple polymer sheets goes through many mistakes and intermediate configurations. In fact, while I now understand that the assembly of two sheets is fairly well controlled (with the addition of new important figure R19), with the capillary forces imposed by multiple polymer sheets the interactions become extremely complex and there can be many local energy minima that trap the systems in metastable configurations.

d) I agree with the authors that the use of light is more desirable than the control via temperature changes.

In all, I could recommend publishing this paper if some claims of the paper are reduced and if further emphasis is placed on the formation of the 3D structures across multiple layers. Here I provide a detailed list of comments.

1) The authors should describe in more detail the limitation of the technique. For example, the word "repeatability" is mentioned many times but I could not find indication of the number of experiments, or relative frequency of different types of assembly. There should be an extensive discussion about this.

2) The authors should be more careful in their use of the word "topological". For example, there is no topological difference between a rectangle with high-aspect ratio formed by the end-to-end assembly and that formed by the side-to-side assembly. There can be, however, a topological difference in the 3D structures where the links are formed across different fluid layers. Therefore the use of "topological" is especially inappropriate in the first part and should be amended.

3) Figure R19 is really crucial in understanding the assembly of the polymer sheets, therefore I greatly recommend moving it to the main paper, and not in the supplemental material.

4) The authors should emphasize from the beginning that the formation of 3D structures is a unique feature of this work, that is only achievable with the ability of their elastomers to achieve positive and negative mean curvature.

5) Minor comments:

- There are some typos and grammar mistakes, such as the use of "flatted" instead of "flattened" (or "flatting" VS "flattening").

- The notes in figure 3 are too small and unreadable.

- There are too many supplemental videos (20). Some of them are not so helpful and should be either grouped into fewer videos (e.g. 12, 13, 17; 18-19) or deleted (e.g. 16).

Reviewer #3 (Remarks to the Author):

I believe the authors have adequately addressed my prior concerns as well as those of the other reviewers.

Response to referee 1

Comments 1:

the authors have answered very extensively to the point raised by this reviewer and have provided additional information that are needed to understand the physics of the assembly. The authors have also spend considerable effort to improve certain paper material including videos. It can now be published, and it will hopefully raise interest in the community to give more feedback.

Response to referee:

We highly appreciate the positive comments and recommendation from the reviewer.

Response to referee 2

Comments 1:

I am reviewing the amended version of the manuscript "Optocapillarity-driven assembly and reconfiguration of liquid crystal polymer actuators" by Hu et al. I would like to reiterate that the experiments are indeed very impressive and that the work is worth publishing. I read from the journal scope that "papers published by the journal aim to represent important advances of significance to specialists within each field". I still claim that the important advance of this paper is the formation of structures across multiple layers. I would like to respond to the authors' answers in detail.

Response to referee:

We highly appreciate the professional comments from the reviewer.

Comments 2:

a) The effect shown by the LC elastomers was demonstrated in hydrogels as the authors mention. The objection of the authors is that the prescribed shape of the hydrogel is decided in advance by lithography. I argue that the alignment of the LC elastomer has the same role, as it decides how and in which direction the elastomer will bend with light. The difference is that it can be sometimes bent in two opposite directions.

Response to referee:

We agree with the reviewer's point of view. The fact that the LCP actuators can be flattened and bent in opposite directions, as recognized by the reviewer, is essential for active manipulation of individual components and reconfiguration of the assembly.

Comments 3:

b) Moreover, the authors emphasize their control on topology and I argue that that is mostly not the case. Almost all the assembly the authors show have the same topology, which is a mis-representation throughout the text, as I mention also below.

Response to referee:

We appreciate the comment from the reviewer. To avoid possible confusion to the readers, we have rephrased the texts and changed "topology" to "pattern".

Comments 4:

c) I find the "tetris" experiment interesting. However, it is very clear from the video that in order to get to those structure the assembly of multiple polymer sheets goes through many mistakes and intermediate configurations. In fact, while I now understand that the assembly of two sheets is fairly well controlled (with the addition of new important figure R19), with the capillary forces imposed by multiple polymer sheets the interactions become extremely complex and there can be many local energy minima that trap the systems in metastable configurations.

Response to referee:

We agree with the reviewer that there may exist some local energy minima that can trap the systems in metastable configurations. If more sheets were involved in the assembling system, there may also exist multiple routes with different intermediate states to get to a final desired configuration, which have been discussed in the response of comment 7.

Comments 5:

d) I agree with the authors that the use of light is more desirable than the control via temperature changes.

Response to referee:

Thank you for your comments.

Comments 6:

In all, I could recommend publishing this paper if some claims of the paper are reduced and if further emphasis is placed on the formation of the 3D structures across multiple layers. Here I provide a detailed list of comments.

Response to referee:

We thank the reviewer for the recommendation and the constructive advices. We have added the content emphasizing the formation of the 3D structures across multiple layers in the abstract and the conclusion.

Comments 7:

1) The authors should describe in more detail the limitation of the technique. For example, the word "repeatability" is mentioned many times but I could not find indication of the number of experiments, or relative frequency of different types of assembly. There should be an extensive discussion about this.

Response to referee:

Thank you for your constructive advice. The transformation of the pattern between any two structures is reversible and repeatable. For an example, as shown in the Supplementary Movie for the review, the assembly composed of four actuators shows 5 cycles of repeatable switching between the pattern 1 and pattern 8 (Supplementary Fig. 19c). The number of the switching cycle is theoretically unlimited as long as the photodeformation performance of the LCN actuator is maintained.

Upon reconfiguring assembling patterns, the number of the steps for the transformation between different two structures are varied, which mainly affected by two factors, namely, the degree of transformation and the number of LCP actuators in the assembling structure. The degree of transformation means the degree of the difference between the initial pattern of the assembling structure and the target pattern of the reconfigured structure. The greater the difference, the higher the transformation degree and many more steps must be involved to achieve transformation. For example, for the assembling structure composed of 4 actuators, the degree of transformation from pattern 1 to pattern 6 is relatively low (Supplementary Fig.19c). Because the different between them is that only two actuators at the end of the structure have different orientation. And two steps to change the orientation of the actuators at the end of the structure 1 is enough to achieve the transformation, which only take ~ 7 s (Part 2 in Supplementary Movie 7). Whereas for the transformation from pattern 8 to pattern 23 (Supplementary Fig. 19c), the pattern difference

between these two structures is relatively high. And it will require multiple steps and additionally many parallel paths with different number of transformation steps exist in this complex transformation. As shown in Part 3 of Supplementary Movie 7, five different parallel paths composed of 3 to 5 steps take ~ 62 s (average time) to complete the reconfiguration. In other word, this transformation will require more optical operations than the simple transformation of pattern 1 to pattern 6. Moreover, if many more actuators were involved in the assembly, the number of parallel paths will dramatically increase. Because, as the reviewer' comments, with the capillary forces imposed by multiple polymer actuators, the interactions among actuators become extremely complex and there can be many local energy minima that trap the systems in metastable configurations. It means that more intermediate configurations will emerge, and lead to more optical operations and time required to achieve the target pattern, especially in the case of the high degree of transformation.

Comments 8:

2) The authors should be more careful in their use of the word "topological". For example, there is no topological difference between a rectangle with high-aspect ratio formed by the end-to-end assembly and that formed by the side-to-side assembly. There can be, however, a topological difference in the 3D structures where the links are formed across different fluid layers. Therefore the use of "topological" is especially inappropriate in the first part and should be amended.

Response to referee:

We thank the reviewer for pointing out this important issue. We have rephrased the texts as suggested in the revised manuscript.

Comments 9:

3) Figure R19 is really crucial in understanding the assembly of the polymer sheets, therefore I greatly recommend moving it to the main paper, and not in the supplemental material.

Response to referee:

As suggested, we have moved figure R19 to the main paper as Figure 1d.

Comments 10:

4) The authors should emphasize from the beginning that the formation of 3D structures is a unique feature of this work, that is only achievable with the ability of their elastomers to achieve positive and negative mean curvature.

Response to referee:

We thank the reviewer for the constructive advice. We have revised the abstract to emphasize the formation of the 3D structures across multiple layers. The revised contents in the abstract and the conclusion are shown below:

The revised content in the abstract: *This approach, taking advantage of optocapillarity induced by photodeformation of floating liquid crystal polymer actuators, not only achieves programmable*

and reconfigurable 2D-dimensional assembly, but also uniquely enables formation of three-dimensional structures with tunable architectures and topologies across multiple fluid interfaces.

The revised content in the conclusion: Most intriguing, our optocapillarity-driven system possesses the unique feature that enables the 3D synergistic assemblies at multilayer liquid interfaces, which is only achievable with the ability of photodeformable actuators to achieve positive and negative mean curvature, and have potential application in bioengineering to dynamically construct hierarchical scaffolds for tissue and cell culture.

Comments 11:

5) Minor comments:

- There are some typos and grammar mistakes, such as the use of "flatted" instead of "flattened" (or "flating" VS "flattening").*
- The notes in figure 3 are too small and unreadable.*
- There are too many supplemental videos (20). Some of them are not so helpful and should be either grouped into fewer videos (e.g. 12, 13, 17; 18-19) or deleted (e.g. 16).*

Response to referee:

We are sorry for the typos and mistakes in the earlier version. We have checked the manuscript and corrected the typos and grammar mistakes as much as we can. The note in Fig. 3 has been enlarged in order to improve its readability. We also cut down the numbers of the videos according to the review's suggestion.

Response to referee 3

Comments 1:

I believe the authors have adequately addressed my prior concerns as well as those of the other reviewers.

Response to referee:

We highly appreciate the recommendation from the reviewer.